# Sonic Hedgehog switches on Wnt/planar cell polarity signaling in commissural axon growth cones by reducing levels of Shisa2

**Keisuke Onishi, Yimin Zou\***

Neurobiology Section, Biological Sciences Division, University of California, San Diego, San Diego, United States

**Abstract** Commissural axons switch on responsiveness to Wnt attraction during midline crossing and turn anteriorly only after exiting the floor plate. We report here that Sonic Hedgehog (Shh)-Smoothened signaling downregulates Shisa2, which inhibits the glycosylation and cell surface presentation of Frizzled3 in rodent commissural axon growth cones. Constitutive Shisa2 expression causes randomized turning of post-crossing commissural axons along the anterior–posterior (A–P) axis. Loss of Shisa2 led to precocious anterior turning of commissural axons before or during midline crossing. Post-crossing commissural axon turning is completely randomized along the A–P axis when *Wntless*, which is essential for Wnt secretion, is conditionally knocked out in the floor plate. This regulatory link between Shh and planar cell polarity (PCP) signaling may also occur in other developmental processes.
DOI: https://doi.org/10.7554/eLife.25269.001

## Introduction

The complex neuronal network is in part assembled through the stepwise events of axon pathfinding. Many axonal growth cones follow the instruction of a set of guidance cues until they reach their intermediate targets; they then switch their responsiveness to follow new guidance cues which allow them to navigate to the next intermediate targets (or the final targets). The mechanisms of these important switches are not well understood. In the developing spinal cord, commissural axons arise from the dorsal margin, project along the dorsal–ventral axis and then turn anteriorly after they have reached and crossed the ventral midline (*Bovolenta and Dodd, 1990*). Wnts are expressed in a decreasing anterior-to-posterior gradient in the floor plate and direct the post-crossing commissural axons to turn anteriorly after midline crossing (*Lyuksyutova et al., 2003*). Previous work showed that components of the planar cell polarity (PCP) signaling pathway mediates Wnt attraction and anterior turning of commissural axons (*Lyuksyutova et al., 2003*; *Shafer et al., 2011*; *Zou, 2012*; *Onishi et al., 2013*; *Onishi et al., 2014*). We report here that Shh–Smoothened (Smo) signaling activates PCP signaling by inhibiting the expression of Shisa2, an inhibitor of Wnt signaling that blocks Frizzled3 (Fzd3) glycosylation and cell-surface presentation.

## Results

### Increased Shisa2 expression in dorsal commissural neurons in Smoothened (Smo) conditional knockout mice

As Shh is a major signaling molecule in the ventral midline, we hypothesized that Shh signaling may regulate the expression of genes relevant to the switch of responsiveness of commissural neurons during midline crossing. We, therefore, performed transcriptome analyses in the dorsal spinal cord, where the cell bodies of commissural neurons reside, from control $Smo^{fl/fl}$ and $Smo$ conditional KO

\*For correspondence:
yzou@ucsd.edu

**Competing interests:** The authors declare that no competing interests exist.

(cKO) E11.5 embryos crossed with *Wnt1-Cre*. Consistent with a previous study, we observed compromised anterior turning of commissural axons in *Smo* cKO (*Figure 1—figure supplement 1*) (*Yam et al., 2012*). Dorsal margins of the spinal cord were dissected (50–100 µm from the dorsal edge) and lysed to extract total RNA. Stranded-mRNAs were prepared and sequenced by single-end RNA-sequencing (*Figure 1A*). Among a total of 15,737 identified genes, 363 genes showed significant differences in mRNA levels (*Figure 1B*). The expression of several functional groups related to neuronal differentiation, axon morphogenesis and synaptic function was found to be changed significantly (*Figure 1C*). We confirmed that *Smo* mRNA was indeed significantly reduced in *Smo* cKO (*Figure 1F*). We then compared the expression levels of all core PCP genes, such as *Fzd3*, *Celsr3* and *Vangl2*, that are required for A–P guidance of post-crossing commissural axons (*Figure 1D*) (*Shafer et al., 2011*; *Onishi et al., 2013*). There was no significant difference in mRNA levels between control *Smo^{fl/fl}* and *Smo* cKO. Interestingly, a homolog of a Wnt inhibitor, *Shisa2*, showed a 3.68-fold increase in *Smo* cKO (*Figure 1E*) (*Yamamoto et al., 2005*).

We next tested whether *Shisa2* mRNA level is increased in the dorsal spinal cord commissural neurons in *Smo* cKO using in situ hybridization (*Figure 2A,B*). To localize commissural neuron cell bodies, we performed combined fluorescent in situ hybridization and immunohistochemistry with a dI1 neuronal marker, Lhx2. Lhx2 is expressed in dI1 neurons derived from Atoh1-positive pdI1 progenitors. The dI1 neurons, which express high levels of Lhx2 (Lhx2$^{high}$) at E11.5, project axons contralaterally (dI1c), whereas the Lhx2$^{low}$ (and Lhx9$^{high}$) dI1 neurons, which are located ventral to the Lhx2$^{high}$ neurons at E11.5, project axons ipsilaterally (dI1i) (*Figure 2A*) (*Ding et al., 2012*). The axons of Lhx2$^{high}$ dI1 neurons reach and cross the floor plate at E11.5, whereas the Lhx2$^{low}$ dI1 neurons are a later population and extend axons ventrally at E12.5. We found that *Shisa2* mRNA level was low in the control *Smo^{fl/fl}* dorsal spinal cord but increased in the Lhx2$^{high}$ dI1 neurons in *Smo* cKO (*Figure 2B,C*). These results suggest that the *Shisa2* expression in dI1c commissural neurons is regulated by Shh–Smo signaling during midline crossing. We noticed that, in *Smo* cKO, a few Lhx2$^{low}$ neurons dorsal to the Lhx2$^{high}$ dI1 neurons also showed upregulated levels of *Shisa2* mRNA. These may be later-born Lhx2$^{high}$ dI1 neurons, whose expression of Lhx2 is still usually not yet fully activated.

## Shisa2 inhibits Fzd3 glycosylation and cell surface presentation

Shisa2 is a member of a family of transmembrane proteins with a single transmembrane domain. *Xenopus* Shisa, xShisa, interacts with xFrizzled8 and retains xFrizzled8 in the endoplasmic reticulum (ER), thus inhibiting canonical Wnt signaling (*Yamamoto et al., 2005*). We first tested whether mouse Shisa2 interacts with mouse Fzd3 by co-immunoprecipitation and found that Fzd3 and Shisa2 interact with each other (*Figure 3A,B*). As a control, we also tested interaction between Shisa2 and Insulin receptor β (IRβ), a cell-surface receptor unrelated to Fzd3, and found that IRβ was not co-immunoprecipitated with Shisa2 (*Figure 3—figure supplement 1*). We previously reported that mouse Fzd3 is glycosylated (*Onishi et al., 2013*). When expressed in HEK293 cells, Fzd3 shows three major bands, which correspond to 'phosphorylated and glycosylated', 'glycosylated' and 'unphosphorylated and unglycosylated (unmodified)' proteins (*Figure 3B*). We found that Shisa2 reduced the amount of 'phosphorylated and glycosylated' and 'glycosylated' Fzd3 (*Figure 3A,B*); the major form of Fzd3 that co-immunoprecipitated with Shisa2 is the 'unmodified' form (*Figure 3B*). We then analyzed the level of Fzd3 at the cell surface using a surface biotinylation assay and found that Shisa2 expression robustly decreased the amount of Fzd3 on the cell surface (*Figure 3C,D*); the cell surface level of IRβ is meanwhile unaffected (*Figure 3C,D*). GAPDH is a cytoplasmic marker and IRβ is a cell surface marker, which allow assessment of fractionation. IRβ was detected both in cell surface fraction and in total cell extract, whereas GAPDH was only found in total extract (*Figure 3C,D*). Finally, we performed immunocytochemistry to characterize Fzd3 localization. In the absence of Shisa2, Fzd3–EGFP is localized diffusely in the membrane compartments of the entire cytoplasmic area and on the plasma membrane (*Figure 3E*). In Shisa2–myc-expressing cells, Fzd3–EGFP and Shisa2–myc are colocalized with Calnexin, an ER marker, suggesting that Shisa2 blocks Fzd3–EGFP transport from ER to Golgi and cell surface (*Figure 3E,F*). Therefore, our results suggest that Shisa2 may inhibit Wnt/PCP signaling by inhibiting Fzd3 glycosylation and trafficking to the cell surface.

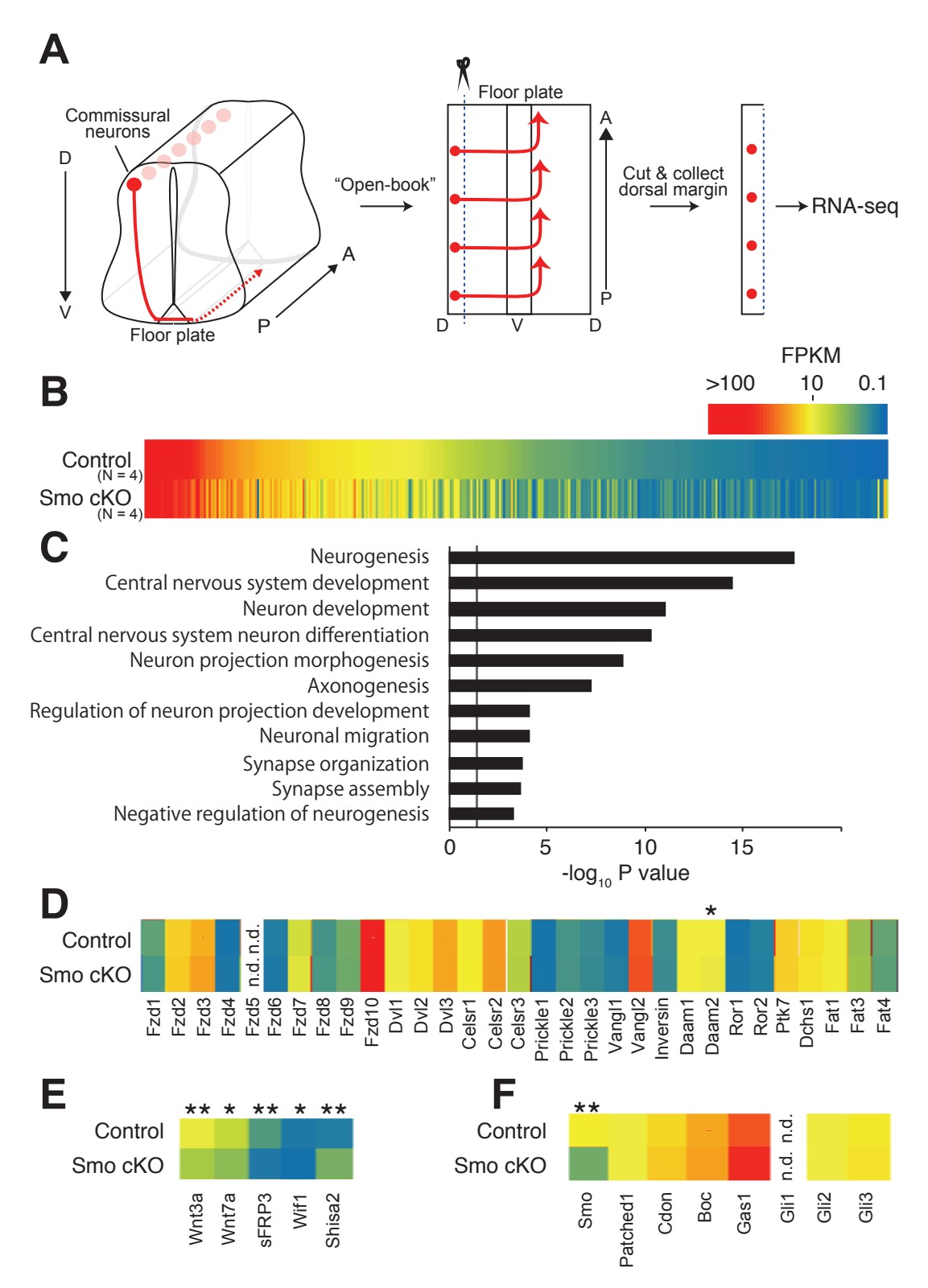

**Figure 1.** Shh–Smo signaling regulates Shisa2 expression in dorsal spinal cord. (**A**) Schematics of dorsal commissural neuron trajectory in mouse spinal cord at E11.5. Cell bodies of commissural neurons (red circles) are close to the dorsal margin of the spinal cord. Their axons (red line) project to the ventral midline and cross to the contralateral side of the spinal cord (dashed red line). Using an 'Open-Book' preparation, dorsal margins from $Smo^{fl/fl}$ (control) or $Smo^{fl/fl}$; $Wnt1$-Cre ($Smo$ cKO) were isolated followed by RNA extraction and RNA-sequencing. (**B**) Heat map of changes of mRNAs levels. (**C**)
*Figure 1 continued on next page*

*Figure 1 continued*

Gene ontology analysis. (D) Heat map of expression levels of PCP genes in control and *Smo* cKO. No significant changes were observed except *Daam2*. (E) Heat map of expression levels of Wnt pathway genes. mRNA levels of *Wnt3a*, *Wnt7a*, *sFRP3* and *Wif1* were decreased in *Smo* cKO. *Shisa2* mRNA level was increased ($FPKM_{control} = 1.25$; $FPKM_{cKO} = 4.82$; q = 0.0038). (F) Heat map of expression levels of Shh pathway genes. No significant changes were observed except for *Smo* itself ($FPKM_{control} = 14.62$; $FPKM_{cKO} = 4.63$; q = 0.0038). * denotes a q-value of = 0.048, ** denotes a q-value of = 0.0039.

DOI: https://doi.org/10.7554/eLife.25269.002

The following figure supplement is available for figure 1:

**Figure supplement 1.** *Smo* is required for A–P guidance of commissural axons.

DOI: https://doi.org/10.7554/eLife.25269.003

## Fzd3 glycosylation is required for its cell surface expression

To further test the importance of the glycosylation of Fzd3 in its trafficking, we set out to identify the glycosylation site(s) of Fzd3. A recent co-crystal structure of xWnt8 and xFz8 revealed a glycosylation site in the cysteine rich domain (CRD) region of xFz8, which is conserved among all vertebrate Frizzle proteins and which may be important for binding to Wnt ligands (*Janda et al., 2012*). In mouse Fzd3, this site is N42 (*Figure 4A*). Fzd3 and Fzd6 have an additional putative glycosylation site (N356 in mouse Fzd3; *Figure 4A*) in their extracellular loop II. To test whether these putative sites are glycosylated, we made two point mutants, N42Q and N356Q, and the double mutant (2NQ [or N42Q + N356Q]). These Fzd3 mutants, N42Q, N356Q and 2NQ, showed faster mobility (*Figure 4B,C*), suggesting that N42 and N356 are indeed glycosylation sites. The cell-surface level of Fzd3 (N42Q) is significantly reduced, whereas Fzd3 (N356) is completely undetectable (*Figure 4D*), suggesting that glycosylation at N356 is more crucial for Fzd3 trafficking than that at N42Q. To better characterize Fzd3 glycosylation, we enriched glycosylated proteins using the Glycoprotein Isolation Kit, WGA (Thermo Scientific) (*Figure 4E*). The wheat germ agglutinin (WGA) preferentially interacts with N-acetyl glucosamine (GluNAC), terminal GluNAC, and sialic acid structures, so that WGA-immobilized agarose allows us to enrich glycoproteins with modified GluNAC or sialic acid. We found that wildtype Fzd3 is detected in the glycoprotein fraction, whereas mutant Fzd3 (2NQ) is absent from the glycoprotein fraction, further confirming that N42 and N356 are glycosylation sites in Fzd3.

To ask whether Shh–Smo signaling regulates Fzd3 glycosylation in vivo, we examined the glycosylation level of endogenous Fzd3 in the dorsal spinal cord in *Smo* cKO. We generated polyclonal anti-Fzd3 antibodies and performed western blot. We detected two bands corresponding to 'glycosylated' and 'unmodified' bands, which are both absent from $Fzd3^{-/-}$ lysate (*Figure 4—figure supplement 1*). However, we noted a non-specific band that runs at the same position as the 'phosphorylation and glycosylation' band (marked with an asterisk). Therefore, this antibody can be utilized to detect 'glycosylation only' and unmodified Fzd3, but not the 'phosphorylation and glycosylation' band. We isolated glycoprotein using the Glycoprotein Isolation Kit, WGA (Thermo Scientific) and performed western blot to detect how much Fzd3 is glycosylated. We found that Fzd3 glycosylation ('glycosylation only' band) was robustly reduced in *Smo* cKO (*Figure 4F*), suggesting that Shh–Smo signaling does indeed regulate Fzd3 glycosylation in vivo.

## Forced expression of Shisa2 lead to A–P guidance defects of post-crossing commissural axons

To test whether Shisa2 downregulation is necessary for proper A–P guidance of commissural axons, we forced expression of Shisa2 using a heterologous CAG promoter in commissural neurons. We co-electroporated Shisa2-expressing plasmids with tdTomato-expressing plasmids into the dorsal spinal cord commissural neurons and analyzed their axon trajectory revealed by tdTomato in an 'open-book' culture (*Figure 5A*) (*Lyuksyutova et al., 2003*; *Wolf et al., 2008*; *Shafer et al., 2011*; *Onishi et al., 2013*). Axons expressing control plasmid extended to the midline, crossed, and turned normally anteriorly (*Figure 5B,C*; 85.9 ± 6.79%). By contrast, we found that axons of Shisa2-expressing neurons turned both anteriorly and posteriorly randomly after midline crossing (*Figure 5B,C*; 54.3 ± 7.89%). This result suggests that the downregulation of Shisa2 is necessary for proper A–P guidance of commissural axons.

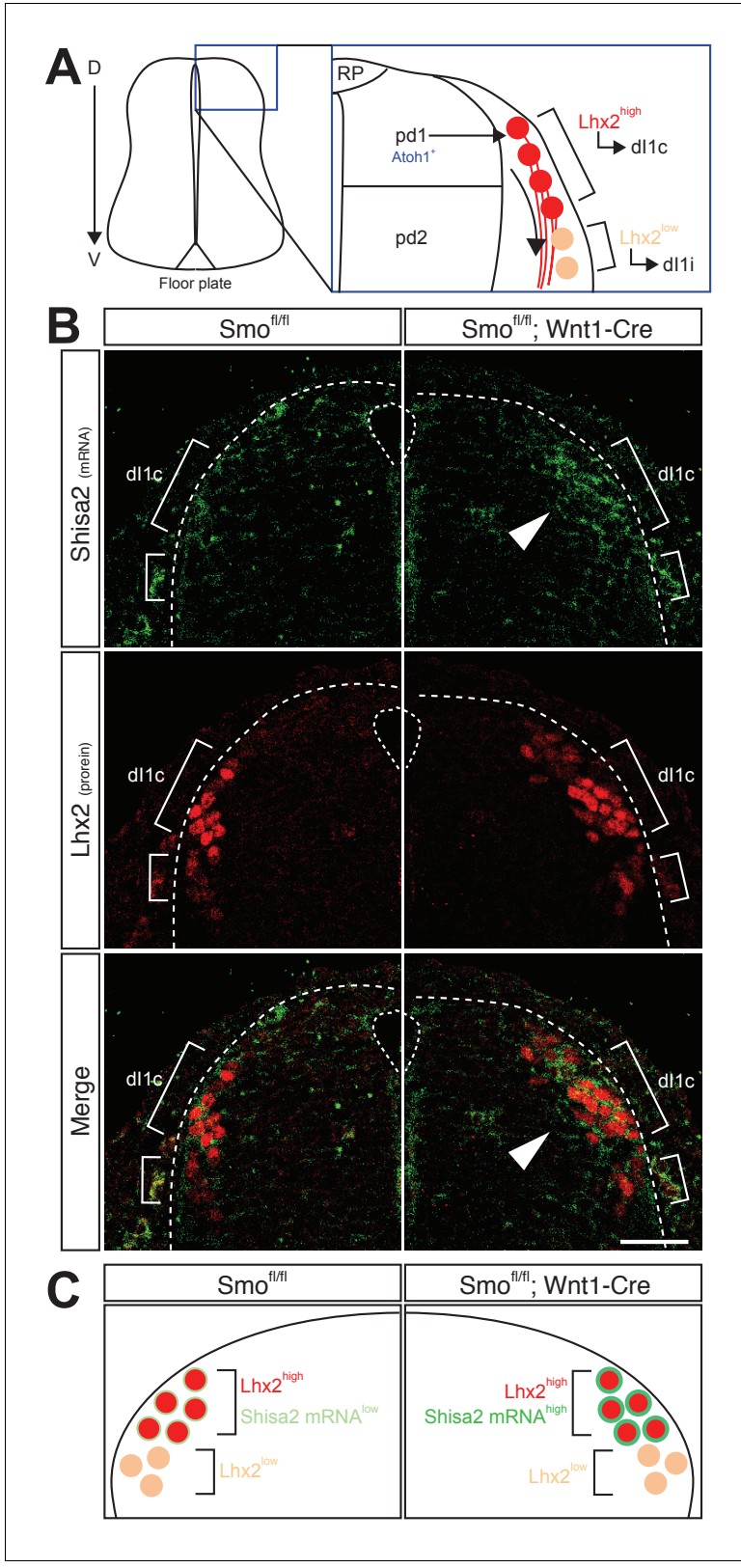

**Figure 2.** Shh–Smo signaling regulates mRNA level of Shisa2 in dl1 commissural neurons. (**A**) A schematics of dl1 commissural neurons in dorsal spinal cord. Atoh1-positive pdl1 progenitors differentiate into dl1 neurons. After differentiation, cell bodies migrate laterally. Lhx2[high] neurons project axons contralaterally (dl1c), whereas Lhx2[low] neurons migrate more ventrally at E11.5 (dl1i). Lhx2[low] neurons start to extend axons at E12.5. (**B**) *Shisa2* mRNA is
*Figure 2 continued on next page*

*Figure 2 continued*

elevated in Lhx2$^{high}$ dI1 neurons in *Smo* cKO dorsal spinal cord (white arrow head). Dash lines indicate the lateral margin of spinal cord. Scale bar: 50 μm.

DOI: https://doi.org/10.7554/eLife.25269.004

As xShisa also binds to FGFR and as our RNA-seq data suggest that FGFR1, FGFR2 and FGFR3 are expressed in dorsal spinal cord (*Figure 5—figure supplement 1A*) (*Yamamoto et al., 2005*), we tested whether FGFRs are required for A–P guidance of commissural axons by blocking FGFR activity using pharmacological inhibitor LY2874455 (inhibits FGFR1, FGFR2, FGFR3, FGFR four and VEGFR2; *Figure 5—figure supplement 1B*). First, we tested whether LY2874455 can abolish FGFR autophosphorylation in our 'open-book' explants and found that LY2874455 suppressed phosphor-Y653/654 significantly (*Figure 5—figure supplement 1C*; Vehicle (DMSO), 1 (normalized); LY2874455, 0.43 ± 0.05). We then found that LY2874455 did not affect A–P guidance of commissural axons (*Figure 5—figure supplement 1D,E*), suggesting that FGFR activity is not required for anterior turning of post-crossing commissural axons.

## Knocking down Shisa2 caused precocious anterior turning of precrossing axons through activation of Wnt-PCP signaling

To investigate whether Shisa2 prevents pre-crossing commissural axons from responding to Wnts, we developed shRNA constructs against Shisa2 (*Figure 6A,B*). As a control, we utilized scrambled shRNA (*Shafer et al., 2011*; *Onishi et al., 2013*). We then electroporated control and Shisa2 shRNAs into dorsal spinal cord. We found that the majority of control-shRNA-expressing commissural axons turned anteriorly after crossing the floor plate (*Figure 6C,D*; 84.3 ± 2.3%), while a small subset of axons turned before crossing (12.6 ± 2.3%) or in the floor plate (3.0 ± 1.0%). This is consistent with our previous finding using this 'open-book' explant culture system (*Wolf et al., 2008*). By contrast, the expressino of both Shisa2 shRNAs significantly elevated the proportion of commissural axons that turned before crossing and in the floor plate (*Figure 6C,D*) and decreased the proportion of those turning after crossing. To control for off-target effects of these shRNAs, we also performed rescue experiments. We introduced four silent mutations in the shRNA#1 targeting site, and two in the #2 site, and confirmed that the expression level of this mutant Shisa2 (which still encodes WT Shisa) is not suppressed by shRNAs (*Figure 6A,B*). We then electroporated Shisa2 shRNAs together with the Shisa2 rescue construct and found that it rescues the phenotype of Shisa2 shRNAs (*Figure 6C,D*).

To test whether the precocious turning in Shisa2 knockdown mutants is caused by the premature activation of Wnt-PCP signaling, we performed double knockdown of Shisa2 and Vangl2, an essential component of Wnt-PCP signaling. We previously reported that Vangl2 is required for Wnt-stimulated axon outgrowth and essential for anterior turning of dorsal commissural axons (*Shafer et al., 2011*). We utilized the same Vangl2 shRNA and confirmed that *Vangl2* knockdown led to A–P guidance defects (*Figure 6—figure supplement 1*). Then, we co-electroporated *Shisa2* shRNA and *Vangl2* shRNA into the dorsal spinal cord, and found that precocious anterior turning does not occur (*Figure 6C,D*), supporting our notion that the precocious turning produced by Shisa2 knockdown is due to the premature activation of Wnt–PCP signaling.

To confirm that Shisa2 regulates Fzd3 trafficking in neurons, we directly tested whether Shisa2 knockdown can change cell surface levels of Fzd3 protein in the growth cone of commissural neurons. We utilized a construct that we developed in a previous study, in which Fzd3 was tagged with tdTomato to the carboxyl domain and the FLAG epitope tag was engineered to the extracellular N-terminus (*Figure 6—figure supplement 2A*) (*Onishi et al., 2013*). We can label the cell surface Fzd3 using anti-Flag antibodies, while the total Fzd3 protein can be detected by the tdTomato signal. In neurons expressing control shRNA, only a small proportion of the Fzd3 protein was on the cell surface (*Figure 6—figure supplement 2B,C*). On the other hand, Shisa2 knockdown led to an increase of Fzd3 protein on the surface of the growth cones (*Figure 6—figure supplement 2B,C*). These results further support the conclusion that Shisa2 regulates Fzd3 cell-surface expression in the growth cones of commissural neurons.

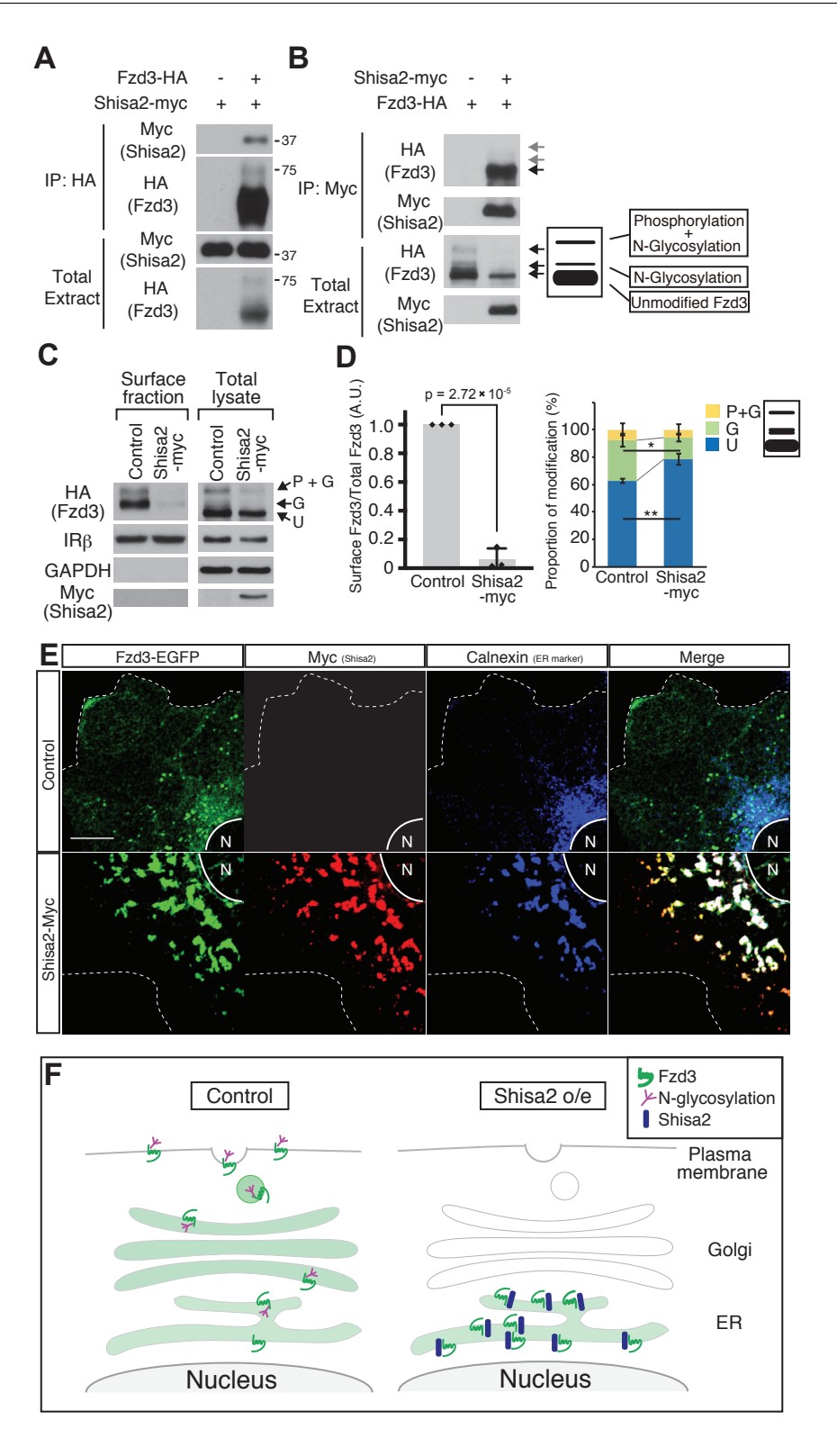

**Figure 3.** Shisa2 inhibits Frizzled3 (Fzd3) glycosylation and translocation to the plasma membrane. (A) Shisa2 was pulled down by Fzd3 in a co-immunoprecipitation assay. (B) Fzd3 was pulled down by Shisa2 in a co-immunoprecipitation assay. The major form of co-immunoprecipitated Fzd3 is the unmodified form. Arrows indicate the bands corresponding to different forms of Fzd3. (C) Fzd3 surface expression and glycosylation are inhibited by Shisa2. Cell surface proteins were labeled with biotin and then precipitated with Neutravidin agarose. Precipitants and total lysates were subject to

*Figure 3 continued on next page*

*Figure 3 continued*

immunoblotting with the indicated antibodies. p+G represents 'phosphorylated and glycosylated', G represents 'glycosylated', and U represents 'unmodified' Fzd3. (**D**) Quantification of cell-surface levels of Fzd3 (left panel) and the proportion of different forms of Fzd3 (right panel). Shisa2 overexpression robustly suppresses cell-surface expression of Fzd3. Shisa2 overexpression significantly increases the amount of 'unmodified' Fzd3 and reduces 'glycosylated' Fzd3. Data are the mean ±SD from four independent experiments. * denotes p<0.05, ** denotes p<0.005. (**E**) Fzd3–EGFP was retained in the ER in the presence of Shisa2. In the control, Fzd3–EGFP was localized widely in the periphery as well as in the perinuclear region, as indicated by Calnexin, an ER marker. By contrast, in Shisa2-overexpressing cells, most Fzd3–EGFP was retrained in the perinuclear region, co-localized with Shisa2 and Calnexin. Scale bar: 10 µm. N: nucleus. Dashed lines indicate cell outlines. (**F**) A schematic of Shisa2 effects on Fzd3 trafficking. Left: in control cells, Fzd3 is glycosylated in ER and translocated to the cell surface through the Golgi/trans-Golgi network. Right: in Shisa2-expressing cells, glycosylation of Fzd3 is inhibited and Fzd3 is not translocated to the cell surface.

DOI: https://doi.org/10.7554/eLife.25269.005

The following figure supplement is available for figure 3:

**Figure supplement 1.** IRβ is not co-immunoprecipitated with Shisa2.

DOI: https://doi.org/10.7554/eLife.25269.006

## Conditional knockout of Wntless in the floor plate lead to the A–P guidance defects of post-crossing commissural axons

Although we have obtained genetic evidence for the essential roles of several PCP components in the A–P guidance of commissural axons, the genetic evidence for a role for Wnts in this guidance is still lacking. This is in part due to the fact that at least five Wnts are expressed in a decreasing anterior-to-posterior gradient (*Lyuksyutova et al., 2003*; *Agalliu et al., 2009*) our unpublished results). As *Wntless* (*Wls*) is essential for the secretion of all Wnt proteins (*Figure 7A*) (*Bänziger et al., 2006*; *Fu et al., 2009*), we knocked out *Wls* specifically in the floor plate by crossing the *Wls* floxed allele with the *Shh-CreER^T2* line; in the resulting line, *Cre* is only expressed in the floor plate and can be activated by intraperitoneal injection of tamoxifen into pregnant females (*Harfe et al., 2004*).

First, we confirmed the specificity of *Shh-CreER^T2* by crossing with *R26R-LSL-tdTom* mice. After administration of tamoxifen at E8.5, embryos were collected at E11.5 and transverse sections were stained with floor plate and ventral spinal cord markers (*Figure 7B*). We observed that tdTomato expression is specifically activated in the notochord and floor plate, where endogenous Shh is expressed (*Figure 7C*). We also confirmed the knockout of the *Wls* gene from the floor plate by in situ hybridization (*Figure 7D*). Second, we tested whether conditional knockout of *Wls* in the floor plate may cause patterning and cell fate defects. At a much later stage, at E13.5, the number of Lhx3⁺; Islet1/2⁺ Median Motor Column (MMC) neurons was found to be reduced, as was that of Lhx3⁻. Islet1/2⁺Hypaxial Motor Column (HMC) neurons were found to be increased in *Wnt4/5* KO (*Agalliu et al., 2009*). We analyzed whether the cell numbers of different motor neuron pools are affected in *Wls* cKO spinal cord at E11.5 (*Figure 7—figure supplement 1A*). We did not observe any differences in the total numbers of motor neurons, marked by Islet1/2, between control and *Wls* cKO mice or the numbers of MMC (Lhx3⁺; Islet1/2⁺) and (HMC + PGC) (Lhx3⁻; Islet1/2⁺) at this early stage (*Figure 7—figure supplement 1B–D*; p>0.05). In addition, the numbers of motor neuron progenitors (pMNs, marked by Oligo2) and V3 interneuron progenitors (p3, marked by Nkx2.2) were also unchanged (*Figure 7—figure supplement 1B–D*; p>0.05).

We then analyzed the midline pathfinding of commissural axons in the *Wls* cKO mice. The isolated spinal cords were prepared as 'open-book', and commissural axons were visualized using iontophoretic injection of DiI at a quarter of the way into the dorsal margin of the spinal cord. In control *Wls^fl/fl* mice (without *Shh-CreER^T2*), commissural axons turn anteriorly after floor plate crossing (*Figure 7E,F*; 85.6 ± 15.7%). On the other hand, in *Wls* cKO mice spinal cords (with *Shh-CreER^T2*), commissural axons turned randomly along the A–P axis after midline crossing (*Figure 7E,F*; 38.7 ± 17.8%), providing genetic evidence that the floor-plate-secreted Wnts are essential A–P guidance cues for post-crossing commissural axons. Due to the intrinsic features of the DiI labeling technique, DiI sometimes over diffuses to label other classes of neurons. Other normal axons tend to have a clear growth pattern and look different from misguided axons in mutants, which tend to show inconsistent wandering patterns. In *Figure 7E*, some neurons, probably located more ventral to the dl1 neurons, were also labeled. Some of them turned before crossing. Other labeled neurons appear to run along the A–P axis close the injection sites. This occurs in both WT control and mutants. These

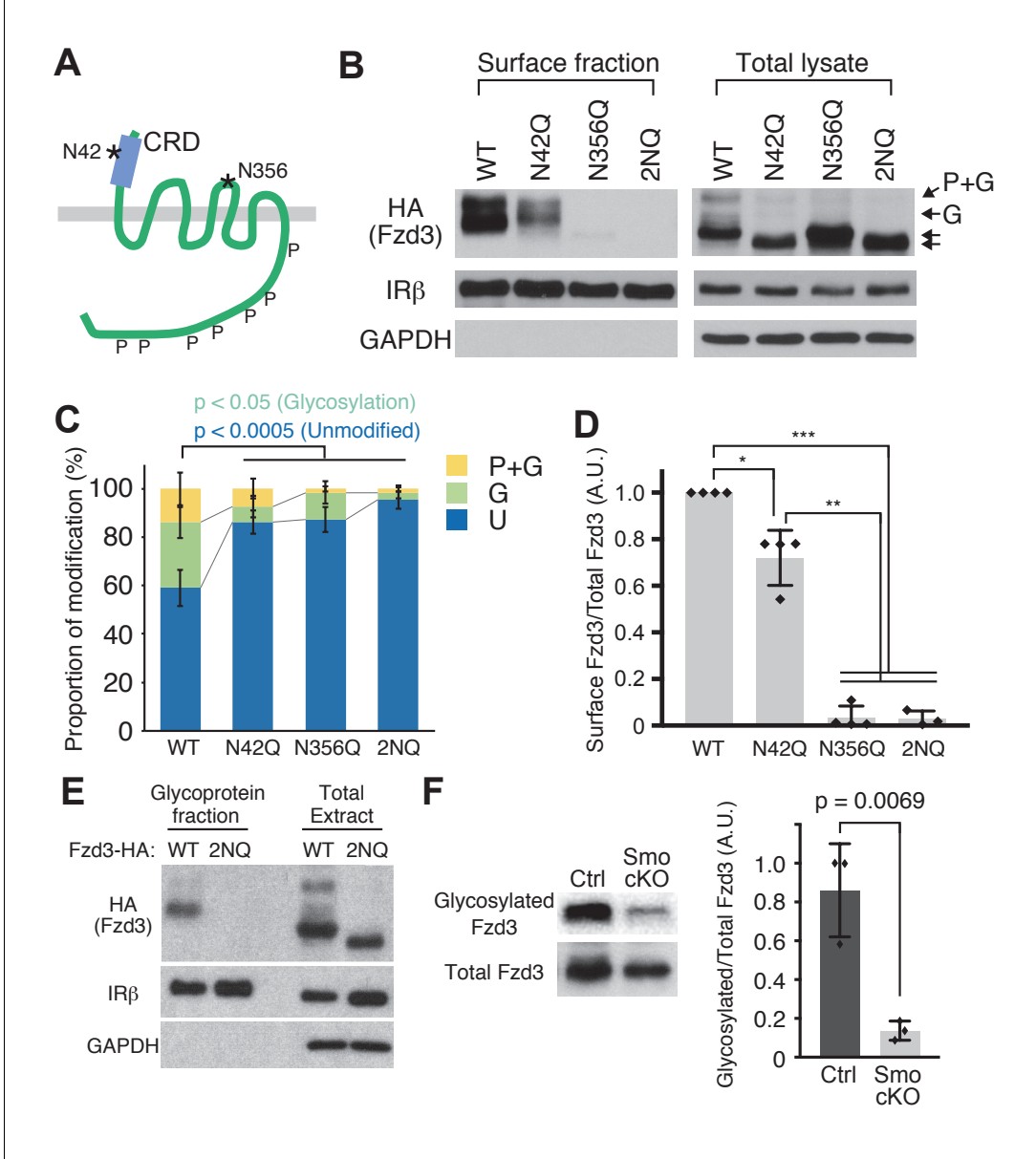

**Figure 4.** Frizzled3 (Fzd3) glycosylation sites required for its cell-surface expression. (**A**) Putative glycosylation sites in Fzd3. N42 is in the cysteine rich domain (CRD), and N356 is in the second extracellular loop. (**B**) N42Q, N356Q and 2NQ mutantations affected Fzd3 glycosylation as well as its cell-surface expression. (**C, D**) Quantification of the cell-surface level of Fzd3 (**C**) and proportions of Fzd3-modification forms (**D**). Single mutations (N42Q, N356Q) and double mutation (2NQ) significantly reduced 'glycosylated' Fzd3 and increased 'unmodified' Fzd3. N42Q mutants showed reduced levels of cell-surface Fzd3. However, N356Q mutation completely eliminated cell-surface Fzd3. Data are the mean ±SD from four independent experiments (2NQ; three experiments). * denotes $p<0.001$; ** denotes $p<5 \times 10^{-7}$; *** denotes $p<1 \times 10^{-8}$. (**E**) Enrichment of glycosylated Fzd3 using the Glycosylation Isolation Kit, WGA (Thermo Scientific). Fzd3 WT was detected in the glycoprotein fraction but 2NQ mutant was not. (**F**) Fzd3 glycosylation in the dorsal spinal cord is reduced in *Smo* cKO. Dorsal spinal cord tissues from control and *Smo* cKO E11.5 embryos were collected and lysed, followed by glycoprotein isolation. Glycosylated Fzd3 was significantly decreased in *Smo* cKO. Data are the mean ±SD from three control and three *Smo* cKO embryos.

DOI: https://doi.org/10.7554/eLife.25269.007

The following figure supplement is available for figure 4:

**Figure supplement 1.** Validation of anti-Fzd3 antibody.

DOI: https://doi.org/10.7554/eLife.25269.008

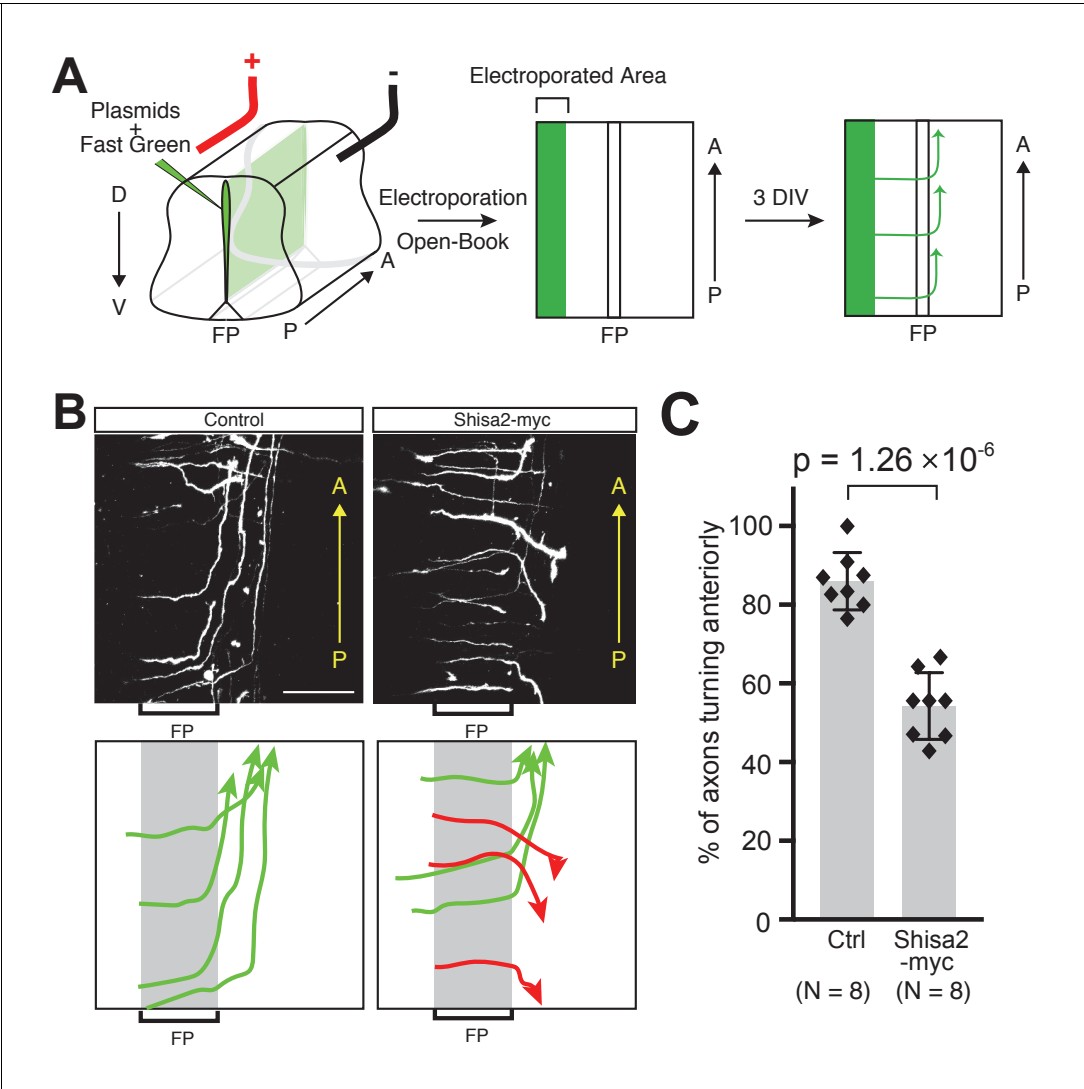

**Figure 5.** Shisa2 overexpression lead to A–P guidance defects after midline crossing. (**A**) Schematic of experimental design. Plasmids cocktail and Fast Green dye were injected into the ventricle of rat embryonic spinal cord (E13) and electroporated into the dorsal region. The spinal cord 'open-book' explants were prepared and cultured in collagen matrix for 72 hr. Axons were visualized by staining for tdTomato expressed by co-electroporated plasmid. (**B**) (Top) Shisa2 expression randomized the turning of post-crossing commissural axons along the A–P axis. (Bottom) Axons in the images on the top panels were traced manually. Green arrows indicate correct (anterior) turning, whereas red arrows indicate posterior turning. FP, floor plate. Scale bars, 50 µm. (**C**) Quantification of A–P guidance defects. The graph represents the percentage of the axons that turned anteriorly (correct turning). Gray bars indicate averages of all data points, black bars indicate standard deviations, and diamond dots indicate individual data points (Control; eight embryos, Shisa2-myc; eight embryos).

DOI: https://doi.org/10.7554/eLife.25269.009

The following figure supplement is available for figure 5:

**Figure supplement 1.** FGFR signaling is not required for A–P guidance of commissural axons.

DOI: https://doi.org/10.7554/eLife.25269.010

are not dIl commissural neurons. But in mutants, post-crossing commissural axons clearly wandered along the A–P axis and show varied patterns of growth.

## Discussion

We found that Shh–Smo signaling reduces *Shisa2* mRNA levels in commissural neurons during or after midline crossing. Shisa2 inhibits Fzd3 trafficking to the cell surface by blocking its glycosylation,

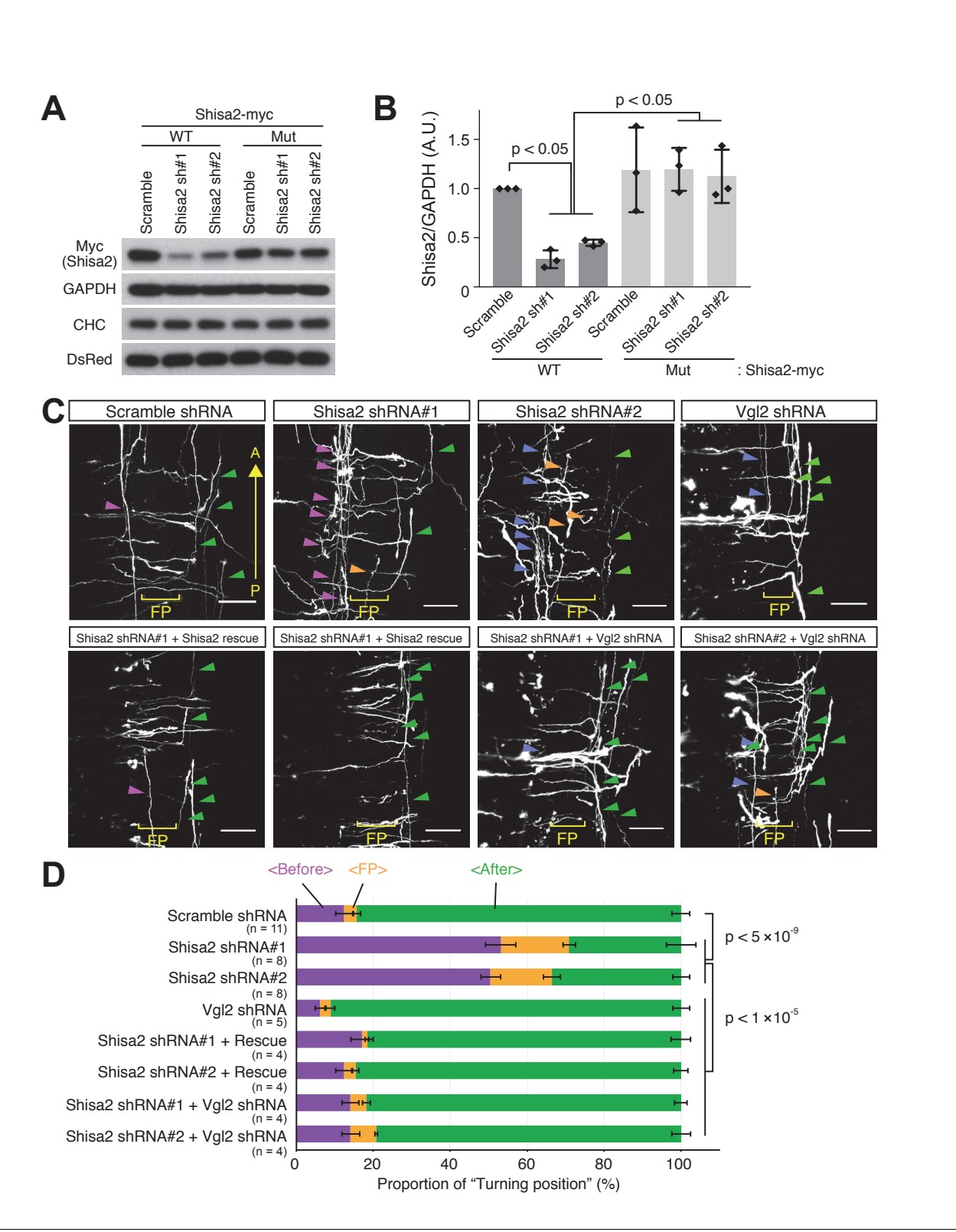

**Figure 6.** Shisa2 knockdown lead to precocious anterior turning before midline crossing. (**A**) Two different shRNA constructs targeting rat Shisa2 can knock down WT Shisa2 but not shRNA-resistant Shisa2 (rescue construct) transfected into 293 T cells. (**B**) Quantification of (**A**). (**C**) Shisa2 knockdown induces precocious turning of commissural axons through premature Wnt–PCP signaling activation. Commissural axons expressing either of the two ShRNA constructs turned before crossing or in the floor plate. This 'precocious turning' was eliminated by co-expression of a Shisa2 rescue construct or

*Figure 6 continued on next page*

*Figure 6 continued*

of Vangl2 shRNA. FP: floor plate. Scale bars: 50 µm. (**D**) The percentage of axons that turn before or after midline crossing is quantified. Purple represents turning before entering the floor plate; orange represents turning in the floor plate; green represents turning after crossing the floor plate. Data are the mean ±SD. n represents the sample number.

DOI: https://doi.org/10.7554/eLife.25269.011

The following figure supplements are available for figure 6:

**Figure supplement 1.** Vangl2 is required for A–P guidance of post-crossing commissural axons.

DOI: https://doi.org/10.7554/eLife.25269.012

**Figure supplement 2.** Shisa2 regulates Frizzled3 (Fzd3) trafficking in the growth cone of commissural neurons.

DOI: https://doi.org/10.7554/eLife.25269.013

keeping Wnt/PCP signaling inactive. Consistent with this, Shisa2 expression using a heterologous promoter led to randomized turning of commissural axons along the A–P axis after midline crossing. Furthermore, Shisa2 knockdown induced Vangl2-dependent precocious anterior turning before midline crossing. At this stage of development, Shh is expressed only in the floor plate and other Hedgehogs are not expressed in or near the spinal cord (*Zhang et al., 2001*). Secreted Shh has two different lipid modifications at its N-terminus and C-terminus (fatty-acid and cholesterol, respectively) that prevent it from diffusing over a long distance (*Sloan et al., 2015*). Therefore, we propose that Shh in the ventral midline is detected by commissural axons and signals retrogradely to inhibit Shisa2 expression in the cell body. This inhibition allows Fzd3 to be trafficked to the cell surface, resulting in the activation of Wnt/PCP signaling in commissural axon growth cones (*Figure 7G*). We also showed here that *Wls*, an essential component of Wnt secretion, is required for A–P guidance of commissural axons, providing genetic evidence for the role of Wnts in A–P guidance of post-crossing commissural axons. This regulatory link between Shh and PCP signaling pathways may also be important in other developmental processes.

The mechanisms for axon responsiveness switches are not fully understood and have attracted increasing attention in recent years. The midline is an excellent model system for addressing this fundamental question. Spinal cord commissural neurons are first guided by Netrin1 to grow from the dorsal spinal cord to the ventral midline, where they cross to the contralateral side (*Kennedy et al., 1994*; *Serafini et al., 1994*; *Serafini et al., 1996*; *Podjaski et al., 2015*; *Yung et al., 2015*). After midline crossing, they lose responsiveness to Netrin1 and become responsive to repellents in the floor plate and the ventral spinal cord, such as Semaphorins and Slits, which turn their trajectory from the dorsal-ventral axis to the anterior-posterior axis (*Zou et al., 2000*). Recent studies suggest that Netrin1 in the ventricular zone is sufficient to guide commissural axons to grow into the floor plate and that floor plate Netrin1 is not required, indicating that Netrin1 functions at a much shorter-range than previously thought (*Dominici et al., 2017*; *Varadarajan et al., 2017*). After midline crossing, commissural axons become responsive to the Wnts, which direct them to turn anteriorly (*Lyuksyutova et al., 2003*). Here, we show that Shh is a midline switch for Wnt/PCP signaling that operates by regulating the mRNA levels of *Shisa2*, which in turn regulates Fzd3 trafficking. We previously observed that it takes approximately 8 hr for commissural axons to cross the midline. This should be sufficient time to downregulate Shisa2 mRNA and protein levels and to allow Fzd3 to be glycosylated and trafficked to the cell surface. This switch mechanism may be optimal for the activation of PCP signaling after midline crossing, given the time it takes to cross the midline. We showed previously that Shh switches on repulsive responses to Semaphorin after midline crossing by inhibiting protein kinase A (PKA) (*Parra and Zou, 2010*). Therefore, Shh is a switch for growth cone responsiveness to both Wnts and Semaphorins. In addition to Shh, other switches in the midline have been proposed. At the *Drosophila* midline, Frazzled/DCC, a Netrin receptor, transcriptionally activates the expression of Commissureless, allowing attraction to be coupled to the downregulation of repulsion in precrossing commissural axons (*Neuhaus-Follini and Bashaw, 2015*). In the vertebrate midline, NrCAM and Gdnf activates Plexin-A1 to switch on responsiveness to Sema3B (*Nawabi et al., 2010*; *Charoy et al., 2012*). Floor-plate-derived neuropilin-2, probably functioning as a Semaphorin sink, was also proposed to be a switch mechanism for Semaphorin responsiveness (*Yang et al., 2009*; *Hernandez-Enriquez et al., 2015*). In rodents, cell-intrinsic signaling proteins, namely the 14-3-3 proteins, were proposed to switch Shh-mediated attraction to repulsion during midline crossing (*Yam et al., 2012*). In chick, Shh binds Glypican 1 to induce the expression of Hhip,

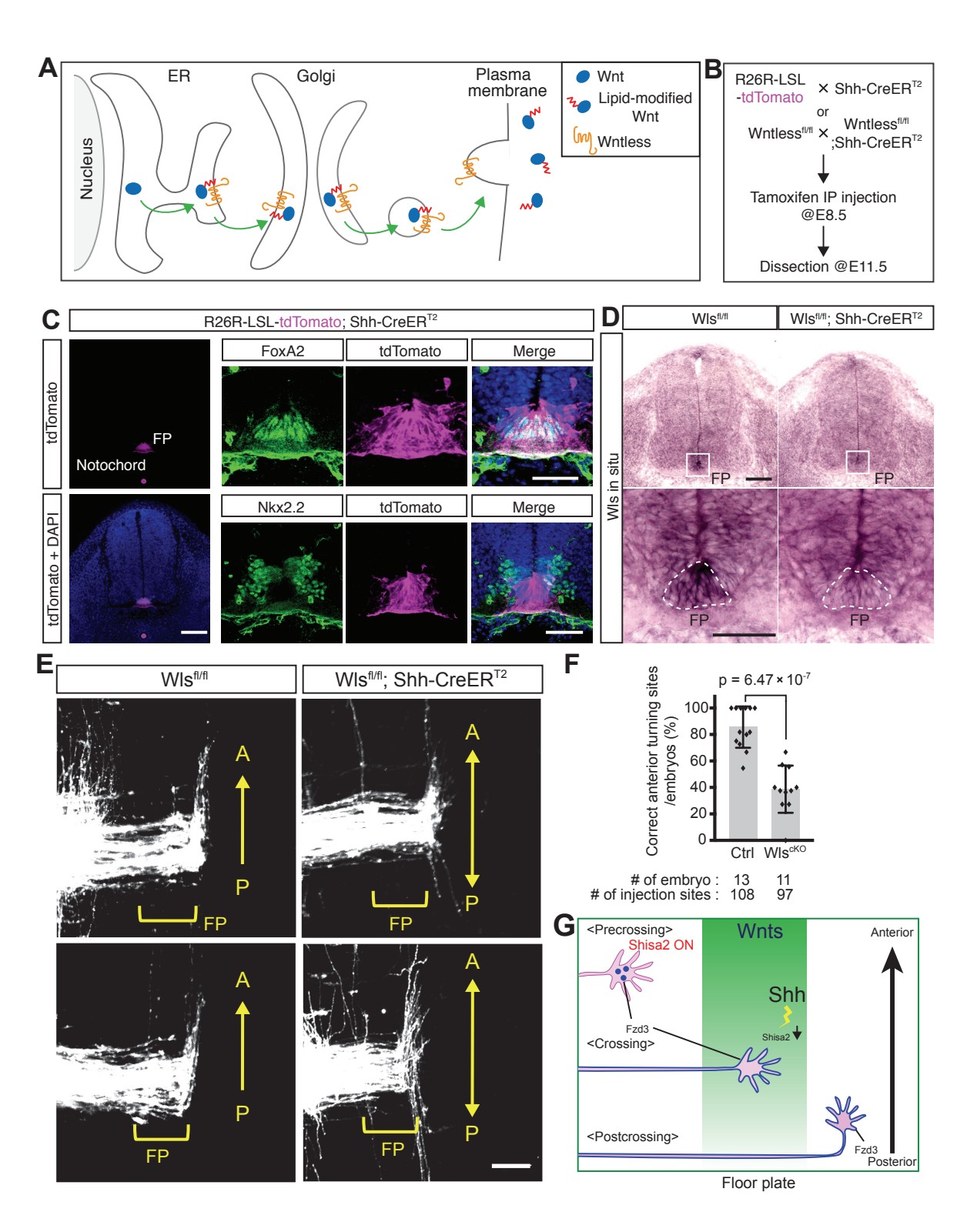

**Figure 7.** *Wntless* (*Wls*) in the floor plate is essential for A–P guidance of commissural axons. (**A**) Schematic of *Wls* function in Wnt secretion. *Wls* is required for Wnts secretion. (**B**) Schematic of experimental design. Tamoxifen (0.1 mg/g of mother) was injected intraperitoneally at E8.5. Embryos were harvested and dissected at E11.5 for immunohistochemistry (IHC), in situ hybridization (ISH) and DiI tracing experiments. (**C**) Cre recombination (tdTomato signal) was detected specifically in the notochord and floor plate. After intraperitoneal injection of tamoxifen (0.1 mg/g mother) at E8.5,
*Figure 7 continued on next page*

*Figure 7 continued*

embryos were collected at E11.5 and transverse sections were immunostained with the indicated antibodies. FoxA2 is a floor plate marker and Nkx2.2 marks the cells immediately outside the floor plate. (D) Loss of *Wls* mRNA in the floor plate of *Wls* cKO (*Wls $^{fl/fl}$*; *Shh-CreER$^{T2}$*). FP, floor plate. Scale bars, 50 µm. (E) Commissural axons, labeled by lipophilic DiI injection into the dorsal spinal cord, showed A–P guidance defects in *Wls$^{fl/fl}$*; *Shh-CreER$^{T2}$*. FP, floor plate. Scale bars, 50 µm. (F) Quantification of A–P guidance defects in (E). The graph represents the percentage of injection sites that showed normal anterior turning (correct turning). Gray bars indicate means of all data points, black bars indicate standard deviations, and diamond dots indicate individual data points (control = 13, cKO = 11). (G) A model for the midline switch of Wnt responsiveness. Before reaching the floor plate, commissural neurons express higher levels of Shisa2. Fzd3 is not glycosylated and not translocated to the cell surface. Therefore, growth cones are not able to sense the Wnt gradient. After commissural axons reach the floor plate, Shh–Smo signaling is activated in the cell bodies of commissural neurons and Shisa2 expression is decreased, allowing Fzd3 to be glycosylated and translocated to the surface of the commissural axon growth cones. After crossing the floor plate, a sufficient amount Fzd3 is on the cell surface and the growth cones can now detect the Wnt gradient and turn anteriorly.
DOI: https://doi.org/10.7554/eLife.25269.014

The following figure supplement is available for figure 7:

**Figure supplement 1.** *Wntless* is not required for cell-fate specification in the ventral spinal cord at E11.5.
DOI: https://doi.org/10.7554/eLife.25269.015

which was proposed to mediate repulsion by Shh (*Bourikas et al., 2005*; *Wilson and Stoeckli, 2013*). Our study here showed that Shh switches on Wnt/PCP signaling by removing an inhibitory mechanism involving Shisa2. Taken together, there may exist highly sophisticated and tightly regulated switch mechanisms to ensure proper changes of responsiveness for axons at intermediate targets. These switch mechanisms can involve gene expression, signal modulation and ligand distribution, which may function together to ensure fidelity.

Although a gene expression regulatory mechanism linking Shh, Tbx2/Tbx3, Shisa3 and canonical Wnt signaling has been observed recently (*Lüdtke et al., 2016*), our study identified a link from Shh to Shisa2 and to non-canonical Wnt/PCP signaling, which regulates cellular and tissue morphogenesis through cytoplasmic signaling. Therefore, Shh may determine the time and space of cell polarity signaling through PCP signaling, as well as by gene expression through the canonical Wnt pathway. Currently, we do not know whether Shisa2 is a direct target of Shh–Smo signaling. It is also possible that the *Shisa2* mRNA level is regulated by a posttranscriptional mechanism.

In this study, we identified two glycosylation sites of Fzd3, one is in the CRD region (N42) and another is in extracellular loop II (N356). It is possible that glycosylation at N42 might be important for the binding affinity with Wnts. Interestingly, the lowest band of the N42Q mutant showed faster mobility than WT (*Figure 4B*), suggesting the possibility that all Fzd3 protein is glycosylated at N42. We show that glycosylation at N356 is more crucial for cell surface expression of Fzd3, this is a unique glycosylation site for Fzd3 and Fzd6, which may also be important for regulating interactions between Fzd3 and other PCP components.

This study also presents the missing genetic evidence for Wnts as axon guidance molecules in vertebrates. Owing to the high redundancy of Wnt family members (there are 19 Wnts in mammals), it has been hard to demonstrate that mutations of individual Wnts lead to axon guidance defects in vertebrates. Here, we conditionally knocked out *Wls* from the floor plate and thus perturbed the secretion of all Wnts from the floor plate. We found that post-crossing commissural axons completely lost directionality along the A–P axis, suggesting that Wnts are the key A–P guidance cues for commissural axons in the ventral midline.

## Materials and methods

### Antibodies and inhibitors

Anti-GAPDH (MAB374; 1/250000; RRID:AB_2107445) was purchased from Millipore. Anti-Nkx2.2 (DSHB, 74.5A5 (sup), 1/500, RRID:AB_531794); anti-TAG1 (DSHB, 4D7 (sup), 1/50, RRID:AB_531775); anti-Olig2 (Millipore, 211F1.1, 1/2000, RRID:AB_10807410); anti-DsRed (Clontech, #632496, 1/2000, RRID:AB_10013483); anti-HA.11 (Covance, 16B12, 1/5000, RRID:AB_291259); anti-Myc (Santa Cruz, 9E10, 1/2000, RRID:AB_627268); anti-Insulin Receptor β (Santa Cruz, sc-711, 1/2000, RRID:AB_631835); anti-Lhx2 (Santa Cruz, sc-19344, RRID:AB_2135660); anti-GFP (Abcam, ab6673, 1/2000, RRID:AB_305643); anti-Calnexin (Abcam, ab22595, 1/2000, RRID:_AB2069006); anti-phospho-FGFR (Y653/Y654, Cell Signaling, #3471, 1/2000, RRID:AB_331072); anti-FGFR1 (Cell

Signaling, #9740, 1/2000, RRID:AB_11178519); and anti-Flag (SIGMA, M2, 1/2000, RRID:AB_439685) were purchased from the indicated venders. Anti-Lhx3 and Islet1/2 antibodies were kind gifts from Dr Pfaff of the Salk Institute. Alexa-conjugated secondary antibodies for mouse/rabbit/rat IgG and mouse IgM were purchased from Molecular Probes (1/500 dilution). HRP-conjugated secondary antibodies for mouse/rabbit/goat IgG were purchased from Jackson ImmunoResearch Laboratories (1/10000 dilution). LY2874455 was purchased from Selleckchem. Anti-Fzd3 rabbit polyclonal antibodies (1/2000 dilution) were generated by the Zou lab. The C-terminal cytoplasmic domain (505–666 a.a.) of mouse Fzd3 fused with glutathione S-transferase (GST) was generated using pGEX4T-1. Recombinant GST–Fzd3cyto protein was expressed in BL21 *Escherichia coli* and purified with glutathione sepharose 4B (GSH beads, GE Healthcare). Anti-Fzd3 polyclonal antibody was generated in rabbits by injection with recombinant GST–Fzd3cyto protein (Covance). Specificity of the anti-Fzd3 antibodies was validated using lysates from Fzd3 knockout dorsal spinal cord (*Figure 4—figure supplement 1*).

## Plasmids

Fzd3–HA, Fzd3–EGFP and tdTomato expressing constructs were described previously (*Shafer et al., 2011*; *Onishi et al., 2013*). Shisa2 cDNA was amplified from the mouse E16.5 brain cDNA library (made by K.O.; SuperScriptIII, Invitrogen) and subcloned into pcDNA3.1-MycHis(A) (Invitrogen). After being subcloned into pcDNA3.1 vectors, Shisa2-MycHis was subcloned into pCAGEN, whereby Shisa2 expression is controlled under CAG promoter, and expressed in dorsal spinal cord commissural neurons without any effects from Shh signaling. To introduce N42Q, N356Q and 2NQ mutation in a Fzd3 background, site-directed mutagenesis against Fzd3 was performed using a QuickChange site-directed mutagenesis kit (Stratagene). Rat Shisa2 cDNA was amplified from a rat E13 dorsal spinal cord cDNA library (made by K.O.; SuperScriptIII, Invitrogen) and subcloned into pcDNA3.1-MycHis (A) (Invitrogen). After being subcloned into pcDNA3.1 vectors, Shisa2-MycHis was subcloned into pCAGEN.

The sequences of the shRNA constructs were follows: control scramble shRNA (5′- GAAACG-GAAAGCAGGTACG −3′), rat Shisa2 shRNA#1 (5′- ACCGCCTGATGGAGACCAT −3′), and rat Shisa2 shRNA#2 (5′- CCACAAATTTCTCTGTACT −3′). Complementary oligonucleotides were annealed and inserted into pSuper vector (Oligoengine). To develop shRNA-resistant rShisa2 expression plasmids, we introduced four and two silent mutations in each target region, respectively. Mutant sequence in shRNA#1 is 5′- ACCGgCTcATGGAaACgAT −3′. Mutant sequence in shRNA#2 is 5′- CCACAAAcTTCTCaGTACT −3′. Lower cases represent mutated sites. Mutations were introduced using QuickChange site-directed mutagenesis kit. All constructs were verified by sequencing (Eton Biosciences, Inc).

## Cell lines and transfection

COS-7 (RRID:CVCL_0224) cells and HEK293T cells (RRID:CVCL_0063) were purchased from ATCC and maintained in Dulbecco's modified Eagle's medium (DMEM) containing 10% fetal bovine serum. Transfection of both COS-7 and HEK293T cells was carried out using 1 mg/ml Polyethyleneimine MAX (Polyscience). Mycoplasma contamination was monitored by DAPI staining. These cell lines are not on the lists of Cross-contaminated or Misidentified Cell Lines from the International Cell Line Authentication Committee.

## RNA isolation and RNA-seq

Control *Smo^{fl/fl}* or *Smo* cKO mouse E11.5 embryos of either sex were dissected. Isolated spinal cords were opened from the dorsal midline ('open-book'). The dorsal margins where cell bodies of dorsal commissural neurons are located were dissected by L-shaped tungsten needle (50–100 μm from the dorsal edge) and collected into microcentrifuge tubes. Total RNA was extracted using RNeasy Plus Micro Kit (QIAGEN). We dissected a total of 14 embryos (seven for each genotype). Total RNA quality was assessed using an Agilent Tapestation. Samples had an RNA Integrity Number (RIN) of greater than 8.0. RNA libraries were generated using Illumina's TruSeq Stranded mRNA Sample Prep Kit using 300 ng of total RNA. RNA libraries were multiplexed and sequenced with 100 base pair (bp) paired single end reads (SR100) to an average depth of approximately 35 million reads per sample on an Illumina HiSeq2500 using V4 chemistry. Sequence results were analyzed

using BaseSpace Cloud Computing System (Illumina). RNA reads were aligned using TopHat Aligment App, followed by Cufflinks Assembly and DE App to determine mRNA expression level. RNA levels were quantified using FPKM (Fragment per kilobase of exon per million fragments mapped). Gene Ontology analysis was performed using GO term mapper (http://go.princeton.edu/cgi-bin/GOTermMapper).

## In situ hybridization combined with immunohistochemistry

For *Shisa2* and *Wls* mRNA, 1406–2224 bp and 414–947 bp regions, respectively, were amplified with PCR and subcloned into pCRII-TOPO vectors (Life Technologies). DIG-labeled antisense probes against each mRNA were transcribed by SP6 RNA polymerase (Roche), whereas sense probes were transcribed by T7 RNA polymerase (NEB). E11.5 mouse transverse sections were prepared on glass slides (Fisher Scientific; #12-550-15). Sections were postfixed with 4% PFA for 5 min, followed by 1 µg/ml Proteinase K treatment and acetylation. Hybridization was performed at 65°C for 20 hr, followed by washes with 0.2 × SSC at 65°C (30 min × 3 times). After SSC wash, RNA probe-hybridized sections were incubated with 10% Donkey serum/TBS for 1 hr for blocking, followed by primary antibody incubation (anti-Lhx2 (1/500) in 3% donkey serum +3% BSA/TBST (TBS +0.1% Tween-20)) at 4°C overnight. After primary antibody wash, sections were incubated with AlexaFluor 647-conjugated anti-goat antibodies (1/500) and Alkaline phosphatase (AP)-conjugated anti-DIG antibodies (Roche/SIGMA, 1/5000, RRID:AB_514497) in 3% donkey serum +3% BSA/TBST at 4°C overnight. After washing, AP activity is detected by HNPP/Fast Red (Roche/SIGMA; 11758888001). Images were taken using Leica SP5 confocal microscopy immediately after Fluoromount G became dry because Fast Red is quickly diffused in the samples. To detect *Wls* mRNA, sections were blocked with 10% Donkey serum/TBS for 1 hr after 0.2 × SSC wash. After blocking, sections were incubated with AP-conjugated anti-DIG antibodies, followed by NBT/BCIP (Thermo Scientific) color detection. Images were taken using Keyence BZX-700 microscope (UCSD Microscopy Core in School of Medicine).

## Dorsal spinal cord extract

Mouse E11.5 embryos from control *Smo^{fl/fl}* or *Smo* cKO were dissected and isolated spinal cords were prepared as 'open-book'. Dorsal margin of the spinal cord was collected into microcentrifuge tubes and then lysed with RIPA buffer (20 mM Tris HCl [pH 7.5], 150 mM NaCl, 1 mM EDTA, 1 mM EGTA, 5 mM NaF, 10 mM β-glycerophosphate, 1 mM $Na_3VO_4$, 1 mM DTT and protease inhibitor cocktail (SIGMA), 1% TX-100% and 0.1% SDS). Lysates were subject to immunoblotting with anti-Fzd3 and anti-GAPDH antibodies. The band intensity was quantified with ImageJ. Statistical analysis was performed with Prism7 (GraphPad Software).

## Glycoprotein isolation

Isolation of glycoprotein from tissue/cell extracts was performed using a Glycoprotein Isolation Kit, WGA (ThermoScientific). For quantification of Fzd3 glycosylation, we prepared lysates from three control *Smo^{fl/fl}* and three *Smo* cKO embryos. The band intensity was quantified with ImageJ. Statistical analysis was performed using Prism7 (GraphPad Software).

## Surface biotinylation assay

The surface biotinylation and NeutrAvidin pull down methods have been described previously (*Shafer et al., 2011*; *Onishi et al., 2013*). Briefly, 48 hr after transfection with indicated plasmids, HEK293T cells (seeded on 20 µg/ml PDL-coated six-well plate) cells were washed with ice-cold PBS (pH 8.0) three times and incubated with 1 mg/ml Sulfo-NHS-LC-Biotin (ThermoFisher Scientific)/PBS for 2 min at room temperature to initiate the reaction, followed by incubation on ice for 1 hr. After quenching active biotin by washing with ice-cold 100 mM Glycine/PBS twice followed by normal ice-cold PBS, the cell lysates were incubated with NeutrAvidin agarose for 2 hr and then precipitated. For quantification, three independent experiments were performed and the band intensity was quantified with ImageJ. Statistical analysis was performed with Prism7 (GraphPad Software).

## Co-immunoprecipitation

48 hr after transfection with the indicated plasmids, HEK293T cells were lysed with IP buffer (20 mM Tris HCl (pH 7.5), 150 mM NaCl, 1 mM EGTA, 5 mM NaF, 10 mM β-glycerophosphate, 1 mM $Na_3VO_4$, 1 mM DTT and protease inhibitor cocktail (SIGMA), 0.1% TX-100). Lysates were immuno-precipitated with anti-HA or anti-Myc antibodies and with protein A/G agarose (Santa Cruz). Experiments were repeated four times and showed similar results.

## Immunocytochemistry

COS-7 cells were plated on coverslips and cultured overnight. Cells were then transfected with 0.4 μg of total DNA. After 40–48 hr, cells were fixed with 4% PFA for 15 min at room temperature, and blocked and permeabilized with 3% normal donkey serum, 3% BSA and 0.1% Triton X-100 in phos-phate-buffered saline (PBS) for 30 min and then incubated with primary antibodies at 4°C overnight. After three washes with PBS, cells were incubated with secondary antibodies for 30 min at room temperature followed by three washes with PBS and mounted in Fluoromount G.

To quantify Fzd3 cell-surface levels, rat commissural neurons expressing Flag– Fzd3–tdTomato were fixed with 4% PFA for 10 min at 37°C, followed by incubation with anti-Flag antibodies (1/5000) for 1 hr at room temperature. After PBS wash, neurons were permeabilized with 0.1% TX-100 for 3 min and blocked with 3% donkey serum +3% BSA/PBST (PBS + 0.1% Tween-20). To detect tdTomato signal, neurons were incubated with anti-DsRed antibodies (1/2000) at 4°C overnight. Neurons were washed and then incubated with secondary antibodies for 1 hr, followed by mounting. Images were taken using Leica SP5 confocal microscope. Signal intensity was quantified using ImageJ. 12 growth cones with scramble shRNA and 15 growth cones with Shisa2 shRNAs from three independent experiments were analyzed.

## Animals

Mice were housed in an environment with a 12 hr light/dark cycle (7 am to 7 pm). *Wnt1–Cre* (#022501), *Shh-CreER$^{T2}$* (#005623), *Smoothened* floxed (#004526) and *Wntless* floxed (#012888) mice were purchased from The Jackson Laboratory. Genotyping of all animals was done by PCR using genomic DNA prepared from tails. Timed-pregnant wild type Sprague-Dawley rats were pur-chased from Charles River Laboratories International, Inc. Experiments were conducted in accor-dance with the NIH Guide for the Care and Use of Laboratory Animals and approved by the UCSD Animal Subjects Committee (Approved Protocol #; S06219, S06222).

## 'Open-book' preparation and DiI axon labeling

Mouse E11.5 spinal cord 'open-books' were prepared as described previously (*Lyuksyutova et al., 2003*; *Wolf et al., 2008*; *Shafer et al., 2011*; *Onishi et al., 2013*). Mouse spinal cord 'open-books' were immediately fixed with 4% PFA. To visualize the trajectory of commissural axons, DiI labeling was used in the open-book preparation. DiI injection and quantification were performed as described previously (*Lyuksyutova et al., 2003*; *Wolf et al., 2008*; *Shafer et al., 2011*; *Onishi et al., 2013*). The number of injection sites and embryos are indicated in the figures.

## Ex utero electroporation

Rat E13 embryos were eviscerated and the notochord was removed. Using a pulled glass needle, plasmids were injected into the neural tube. Using 5 mm gold-plated electrodes (#45–0115; Harvard Apparatus, South Natick, MA), square-wave current was passed across the dorsal neural tubes using a BTX #ECM 830 electroporator. Electroporation conditions were three pulses, 25 V, 100 ms pulse and 1 s interval. After electroporation, the spinal cords were dissected and 'open-book' explants were cultured in a three-dimensional collagen matrix for 72 hr. Culture media were changed every 24 hr. Explants were then fixed with 4% PFA at 37°C and removed from collagen matrix and blocked with 3% normal donkey serum, 3% bovine serum albumin, and 0.5% Triton X-100 overnight. This was followed by incubation with anti-DsRed antibodies for two nights. Explants were then washed with PBST (PBS + 0.5% TX-100) for 1 hr × 5 times and incubated with secondary antibody anti-rabbit IgG conjugated with biotin overnight. After PBST washes, explants were incubated with Streptavidin-488 overnight followed by multiple washes and mounting in Fluoromount G between two coverslips for microscopic analysis. Images were taken using Leica SP5 confocal microscope. To quantify the effect

of Shisa2 overexpression on commissural axon turning, 116 axons were counted for control, and 98 axons were counted for Shisa2 overexpression from eight different spinal cords (from four different litters). To quantify the effect of Shisa2 knockdown, 336 axons were counted for control (11 embryos), 427 axons were counted for shRNA#1 (eight embryos), 320 axons were counted for shRNA#2 (eight embryos), 92 axons were counted for Vangl2 shRNA (five embryos), 101 axons were counted for shRNA#1 + rescue (four embryos), 92 axons were counted for shRNA#2 + rescue (four embryos), 102 axons were counted for shRNA#1 + Vangl2 shRNA (four embryos) and 82 axons were counted for shRNA#2 + Vangl2 shRNA (four embryos).

## Immunohistochemistry

E11.5 mouse embryos were fixed in 4% PFA for 2 hr on ice. After equilibration with 30% (w/v) sucrose in PBS, the fixed embryos were embedded in OCT compound (SAKURA) and frozen. Transverse sections were prepared with a cryostat (CM3050S, Leica) at a thickness of 20 μm and mounted on glass slides (Superfrost Plus, Fisher Scientific). Slides were washed in PBST (PBS + 0.1% TritonX-100) and incubated in 3% BSA in PBST (blocking solution) for 1 hr at room temperature. Slides were further incubated with primary antibodies diluted in blocking solution overnight at 4°C, washed three times for 10 min each time in PBST and then incubated for 2 hr with secondary antibodies diluted in blocking solution at room temperature. The slides were washed again and mounted using Fluoro-mount G (Southern Biotech). All fluorescence images were taken using a Leica SP5 confocal microscopy. All quantification data were obtained using ImageJ software.

## Statistical analysis

Statistical analysis of multiple comparisons was performed using one-way ANOVA followed by a Tukey-Kramer post-hoc test for multiple comparisons using Prism7 (*Figure 1—figure supplement 1C*, *Figure 4C,D*, *Figure 6B,D*, *Figure 6—figure supplement 2*). To compare two groups, Student's *t* test was performed (two-tailed distribution) using Prism7 (*Figure 3D*, *Figure 4F*, *Figure 5C*, *Figure 5—figure supplement 1C*, *Figure 6—figure supplement 1*, *Figure 7F* and *Figure 7—figure supplement 1D*). In the RNA-seq analysis, q-values (modified p-values) were used to estimate significance (consider $q < 0.05$ as significant).

## Acknowledgements

This project was supported by NINDS RO1 NS047484 and R21 NS NS095615 to YZ and a postdoctoral fellowship from the Japan Society for the Promotion of Sciences to KO. RNA-sequencing was conducted at the IGM Genomics Center, University of California, San Diego, La Jolla, CA. We would like to acknowledge the UCSD School of Medicine Light Microscopy Facility (Grant P30 NS047101) and Jennifer Santini for allowing us to use and assistance with Keyence microscopy.

## Additional information

### Funding

| Funder | Grant reference number | Author |
| --- | --- | --- |
| National Institute of Neurological Disorders and Stroke | NS047484 | Yimin Zou |

The funders had no role in study design, data collection and interpretation, or the decision to submit the work for publication.

### Author contributions

Keisuke Onishi, Data curation, Formal analysis, Investigation, Methodology, Writing—original draft, Writing—review and editing; Yimin Zou, Conceptualization, Resources, Supervision, Funding acquisition, Validation, Visualization, Writing—original draft, Project administration, Writing—review and editing

## Author ORCIDs

Keisuke Onishi http://orcid.org/0000-0001-9838-6332
Yimin Zou http://orcid.org/0000-0002-1092-5547

## Ethics

Animal experimentation: Experiments were conducted in accordance with the NIH Guide for the Care and Use of Laboratory Animals and approved by the UCSD Animal Subjects Committee (Approved Protocol #: S06219, S06222).

## Decision letter and Author response

Decision letter https://doi.org/10.7554/eLife.25269.017
Author response https://doi.org/10.7554/eLife.25269.018

## Additional files

### Supplementary files

• Transparent reporting form
DOI: https://doi.org/10.7554/eLife.25269.016

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
