## [Decision Letter]

Thank you for submitting your article "Sonic Hedgehog is a midline switch for Wnt/planar cell polarity signaling in commissural axon guidance" for consideration by *eLife*. Your article has been reviewed by three peer reviewers and the evaluation has been overseen by a Reviewing Editor and Didier Stainier as the Senior Editor. The reviewers have opted to remain anonymous.

The reviewers have discussed the reviews with one another and the Reviewing Editor has drafted this decision to help you prepare a revised submission.

Summary:

The reviewers and Reviewing Editor agree that your analysis of Wnt and Shh interactions on axon guidance at the spinal cord floor plate is a well-done study on how growth cones switch their response from one cue to another. The introduction of the signaling mechanisms involving a new player, Shisa2, brings together your lab's work on Wnt signaling in A-P projection of commissural neuron axons and sonic hedgehog influences at the ventral midline.

Your detailed analyses address how commissural growth cones become competent to respond to Wnt signalling as they cross the floor plate: Shh signaling down-regulates the expression of Shisa2 as commissural axons cross the floor plate. Shisa down-regulation in turn is needed to allow glycosylation and cell surface presentation of the Wnt receptor Fzd3, thereby rendering growth cones responsive to Wnt. Support for this idea includes that 1) Shisa and Fzd3 physically interact; maintenance of Shisa expression in commissural neurons randomizes the growth of commissural axons along the A-P axis, whereas its inhibition leads to premature turning; and conditional inactivation of floor plate-derived *Wntless*, a molecule responsible for the secretion of Wnt proteins, also randomizes the growth of commissural axons along the A-P axis.

The reviewers' positive comments are offset by requests for a number of analyses and controls that they view would strengthen the claims of the paper. None of these constitute new experiments outside of the scope of the current study, but are viewed as essential.*eLife*

Essential revisions:

1) More careful analysis of the timing and cell type specificity of Shisa2 expression, pre- and during floor plate crossing, since this is a key element of the 'gene expression switch' model:a) A comparison of Shisa2 levels before and after midline crossing would strengthen the role of midline Shh as a switch.b) Double labeling the Shisa2 in situs with a dorsal commissural neuron marker would more convincingly show that Shisa2 is up-regulated specifically in dorsal commissural neurons in the Smo conditional knock out.c) Better evaluation of changes in Fzd glycolsylation, Fzd3 protein localization, and visualization of myc-tagged Shisa2 to evaluate expression are also recommended.

2) Proper controls especially for the RNAi experiments, i.e. to rescue with an RNAi resistant Shisa2.

3) To further substantiate a link between Shisa2 and Shh/Wnt function in the in vivo experiments, knock down of Shisa2 in the *Wntless* knockout, which should suppress the premature anterior turning phenotype.

4) The reviewers struggled with comparing phenotypes with varying disruptions in anterior turning, given the difficulties with standardization of size and locus of DiI injections. The *Wntless* knock-out images are much more striking in terms of axons turning posteriorly than are the Smo knock outs, but the quantification works out to be the same. Please comment.

5) Textually, the paper is clearly written, but the tone and conclusions need to be modified:

– The conclusions are overstated and should more precisely reflect the experimental results.

– The claim that this is the first report of a gene-expression-based switch mechanism by a morphogen in axon guidance at an intermediate target is unfounded. Previous work on response switching at intermediate targets dependent on gene expression changes have been published by the Bashaw, Castellani and Charron labs, and their results compare to the present data.

– The evidence for a "loop" between Wnt-Shh signalling is thin. The signaling seems to be in one direction: Shh-smo signaling represses Shisa2, and Shisa2 prevents Frizzled3 cell surface expression.

– If Wnts are concluded to be the major the A-P guidance cue, complete loss of anterior turning should be expected but this is not the case, and in fact axons would be predicted to stall at the midline. Additional information, independent of Wnt, might be involved and should be acknowledged, especially with respect to the *Wntless* knockdown.

Reviewer # 2:

1) The main question is how Shisa2 is regulated by Shh/Smo signaling. It was suggested that this is a gene-expression-based mechanism. However, without direct evidence, it is not appropriate to say it is gene-expression based. As indicated in the discussion, Shisa2 mRNA levels could be regulated by many different mechanisms, and it may not be a Gli target.

2) The effect of Shisa2 and Shh on Fzd3 glycosylation is interesting. However, the western blot comparing endogenous Fzd3 levels used dorsal spinal cord (Figure 4). Although it reflects the levels of Fzd3 in the cell body of commissural neurons, it may not reflect the levels of Fzd3 in the growth cone. Is it possible to measure the Fzd3 levels in the axon/growth cone of commissural neurons?

3) The format of a reference in subsection “Knocking down Shisa2 causes precocious anterior turning of precrossing axons” is wrong.

Reviewer # 3

Figure 2: It is unclear what the significance is of including the Neuropilin data.

Figure 3: In the co-immunoprecipitation experiments, better negative controls would increase confidence in the interaction. The only controls included are to show that the antibodies to the tags don't non-specifically pull down the target proteins, Ideally the authors could show that Shisa does not bind to some other membrane protein. (3E) There are insets included in the top row of images, but it's unclear why there are insets, and they're not significantly larger than the boxed area they are enlarging.

Figure 4: (E) the 'changes' in ratio of glycosylated versus unglycosylated Fzd in the Smo knock-outs is not convincing. The different forms are very poorly resolved. It would be interesting if the authors could speculate on how they think Shisa is affecting Fzd glycosylation.

Figure 5: Shisa2-myc is overexpressed in commissural neurons. Since Shisa2 is myc-tagged, what does shisa-myc expression look like in these neurons? Do the authors observe any difference in Fzd levels or localization in these conditions?

Figure 6: Shisa2 is knocked down with shRNA. The authors need to show that this phenotype can be rescued by over-expression of an RNAi-resistant Shisa2 construct.

Figure 5 and Figure 6: Can Frizzled3 protein localization be visualized by immunofluorescence? If yes, what does Frizzled3 localization look like in commissural neurons when Shisa2 is over-expressed, or knocked down?

Figure 5—figure supplement 1: a positive control for FGF inhibition is needed.

[Editors' note: further revisions were requested prior to acceptance, as described below.]

Thank you for resubmitting your work entitled "Sonic Hedgehog switches on Wnt/planar cell polarity signaling in commissural axon growth cones by reducing levels of Shisa2" for further consideration at *eLife*. Your revised article has been evaluated by Didier Stainier (Senior editor), a Reviewing editor, and two reviewers.

Your manuscript linking Shh signaling and non-canonical Wnt signaling has been improved. Reviewer 2 was satisfied with your revisions but there are some remaining issues cited by reviewers 1 and 3 in their formal review and the consultation that followed that need to be addressed before acceptance. These are summarized below:

Please note that *eLife* usually doesn't consider multiple revisions especially when it appears that the authors did not strive to address all of the original concerns and/or produce an easily readable manuscript. Needless to say, this will be your last opportunity to fix the manuscript to an acceptable level.

Reviewer 1, on details of experiments in text and figures:

1) Figure 2 shows that in Smo mutants there is an increase of Shisa mRNA in a population of Lhx2-low neurons positioned dorsally to the dl1c population. Do these neurons project ipsilaterally? Why do they increase Shisa expression as well?

2) Figure 5 is not very convincing as it is difficult to determine from where the segments of axons indicated by arrows come from. Can the authors replace this image for a more illustrative one?

3) In Figure 6, there are many green/flag puncta outside of the growth cones with the exception of the control one. Can the authors provide better images? Such strong background staining casts doubts on the real increase of surface Fzd3.

4) In Figure 7, there are several axons that seem to project abnormally at the pre-crossing side of the floor plate. This should be explained

Reviewer 1 and 3, and the Reviewing Editor, have multiple concerns about the text:

1) The manuscript is still extremely poorly written. It suffers from errors of syntax, usage and grammar throughout, and needs to be completely rewritten.

2) There is a pervasive lack scholarly acknowledgement of the literature. Reviewer 3 pointed out in his initial review that the paper is poorly referenced and discussed beyond the authors' own data and their interpretation, yet this has not been fully addressed in the revision:a) The authors argue in point 5 of the response to Reviewer 3's initial comments that the work of Charron, Castellani, and Bashaw did not constitute a genuine gene-expression-based switch mechanism by a morphogen in axon guidance at an intermediate target. We agree with Reviewer 3 and would like to see a more cogent discussion and comparison of the work from these three labs in reference to the present data.b) New, very relevant papers on netrin were not cited: two studies have shown that Netrin is actually required in the ventricular zone, not the floor plate and these two studies should be discussed with reference to the authors' findings. Dominici et al., 2017 and Varadarajan et al., 2017. How might these findings interface with the authors' datac) While it is true that the manuscript provides novel evidence for a link between Shh signaling and non-canonical Wnt signaling, as Reviewer 1 has pointed out, there are many examples for a similar relationship with canonical Wnt signaling and these should be referenced.

3) The conclusions have been largely tempered in the revised manuscript, yet there are remaining places where the claims have not been toned down, with a number of unclear and unsubstantiated claims given special emphasis. Reviewer 3 finds this problem persistent in the revised manuscript, citing an excerpt from the discussion: "In this study, we also noticed that Shh also up regulates the mRNA level of a Semaphorin receptor Neuropilin2 (Figure 1—figure supplement 2), which may further sensitize commissural axon growth cones to Semaphorin. Therefore, Shh is a switch of growth cone responsiveness to both Wnts and Semaphorins." No experimental evidence is reported in support of this other than the change in Npn expression.

---

## [Author Response]

Essential revisions:1) More careful analysis of the timing and cell type specificity of Shisa2 expression, pre- and during floor plate crossing, since this is a key element of the 'gene expression switch' model:a) A comparison of Shisa2 levels before and after midline crossing would strengthen the role of midline Shh as a switch.

This is a very good suggestion to determine the timing of Shisa2 expression. However, we would like to point out the technical difficulty of distinguishing pre-crossing and post-crossing commissural neurons in the spinal cord. The cell bodies of pre- and post-crossing axons are all intermingled together in the dorsal spinal cord. There are no molecular markers for pre- and post-crossing commissural axons. It is not possible to use anatomical tracing for the following reason. It is true that we can label the postsynaptic axons and identify their cell bodies by injecting DiI to the post-crossing segment of the axons on the contralateral side. But we can’t selectively label the pre-crossing axons/cell bodies by injecting DiI on the ipsilateral side of the spinal cord because the post-crossing axons/cell bodies will also be labeled as the axons need to first go through pre-crossing segment. In addition, there is no good Shisa2 antibody to visualize endogenous Shisa2 protein levels. And even if there is a good Shisa2 antibody, DiI will also be diffused away when we do immunostaining, which needs detergent. Although we could not experimentally demonstrate the timing of Shisa2 protein expression, we would like to propose that Shh signaling is a switch of the Shisa2 protein level when commissural axons are crossing the midline: 1) Shisa2 mRNA is increased in the cell bodies of the dl1 commissural neurons (Lhx2-positive) in Smo cKO crossed with Wnt1-Cre (new Figure 2) Shh is expressed only in the floor plate and other Hedgehogs are not expressed around the spinal cord at this stage of development (Zhang et al., 2001); 3) Secreted (cleaved N-terminal) Shh has two different lipid modification at its N-terminal and C-terminal (fatty-acid and cholesterol, respectively). Therefore, Shh is a sticky protein and cannot diffuse over a long distance from floor plate to affect gene expression in the dorsal spinal cord (Sloan et al., 2015). We have now emphasized these points in the Discussion section.

b) Double labeling the Shisa2 in situs with a dorsal commissural neuron marker would more convincingly show that Shisa2 is up-regulated specifically in dorsal commissural neurons in the Smo conditional knock out.

For the suggestion to more carefully analyze the cell type specificity of Shisa2 expression, we were able to use a combination of in situ hybridization for Shisa2 (no Shisa2 antibody available) and immunostaining of Lhx2 (dl1 marker) to demonstrate that Shisa2 gene expression is increased in dl1 neurons in Smo cKO (new Figure 2).

In addition, we also further tested whether the gene expression level of Shisa2 may affect Frizzled3 cell surface presentation as we proposed and found that Shisa2 knockdown did resulted in increased Frizzlzed3 cell surface level (Figure 6—figure supplement 2).

Therefore, we propose Shh is a midline switch, which down regulates Shisa2 expression and in turn, lead to the increased Frizzled3 level on the surface of the growth cone membrane.

c) Better evaluation of changes in Fzd glycolsylation, Fzd3 protein localization, and visualization of myc-tagged Shisa2 to evaluate expression are also recommended.

As the glycosylated Frizzled3 band runs very close to the non-glycosylated band, it is hard to evaluate changes of glycosylation of Frizzled3. We purchased the “Glycoprotein Isolation Kit, WGA” from ThermoFisher to isolate and compare glycosylated Fzd3. And we found that Fzd3 glycosylation is significantly suppressed in Smoothened cKO. Please see new Figure 4. To better visualize Fzd3 and Shisa2 localization (in old Figure 3), we prepared larger images to show their localization more clearly. Please see new Figure 3.

2) Proper controls especially for the RNAi experiments, i.e. to rescue with an RNAi resistant Shisa2.

In order to avoid off-target artifacts, we used two different Shisa2 shRNAs and found that they gave rise to very similar phenotypes, suggesting that this phenotype is not because of off-target effect. To further eliminate the possibility of off-target effect, we developed Shisa2 rescue construct, which is insensitive to both shRNA#1 and #2. Then we electroporated shRNA-expressing vector and rescue vector into dorsal spinal cord to see how commissural axons turn. We found that Shisa2 rescue construct restored the precocious turning phenotype of Shisa2 knock down, suggesting that Shisa2 knock down phenotype is due to the reduction of Shisa2 expression level. Please see new Figure 6.

3) To further substantiate a link between Shisa2 and Shh/Wnt function in the in vivo experiments, knock down of Shisa2 in the Wntless knockout, which should suppress the premature anterior turning phenotype.

This would be an excellent experiment. However, due to the limited number of *Wntless* mice we have this experiment would take many more months to finish. To test whether Shisa2 knockdown leas to premature activation of the Wnt/PCP signaling in commissural axons, we decided to also knockdown a key PCP component, Vangl2, as we have shown before that Vangl2 is required for anterior turning of commissural axons (Shafer et al., 2011). We found that double knock down indeed restored the precocious turning phenotype by Shisa2 knock down. This result further supports our working hypothesis. Please see new Figure 6.

4) The reviewers struggled with comparing phenotypes with varying disruptions in anterior turning, given the difficulties with standardization of size and locus of DiI injections. The Wntless knock-out images are much more striking in terms of axons turning posteriorly than are the Smo knock outs, but the quantification works out to be the same. Please comment.

We would like to clarify that we quantify these phenotypes by scoring the percentage of injections sites of DiI whether these sites display normal anterior turning. The axons in Smo cKO appear shorter, probably because this spinal cord was younger and the axons did not have enough time to grow along the anterior-posterior axis after midline crossing.

5) Textually, the paper is clearly written, but the tone and conclusions need to be modified:– The conclusions are overstated and should more precisely reflect the experimental results.

We modified the conclusions to more precisely reflect the results.

– The claim that this is the first report of a gene-expression-based switch mechanism by a morphogen in axon guidance at an intermediate target is unfounded. Previous work on response switching at intermediate targets dependent on gene expression changes have been published by the Bashaw, Castellani and Charron labs, and their results compare to the present data.

We appreciate the reviewer bringing this up. Bashaw lab did show that Frazzled/DCC regulates Comm transcription in a netrin-independent manner. “A Frazzled:DCC-dependent Transcriptional Switch Regulates Midline Axon Guidance” from Dr. Bashaw (Science 2009). So our paper is indeed not the first gene-expression based switch, although it is the first morphogen triggered gene expression switch. We will modify the conclusion to the first Shh morphogen triggered mRNA level change.

We would like to clarify that Castellani did not show gene-expression based switch. They showed proteolysis of PlexinA1 is inhibited by NrCAM and GDNF secreted from floor plate, turning on Sema-Plexin signaling after floor plate.

We would also like to clarify that Charron showed expression level of some 14-3-3 proteins changes during in vitro culture (2 DIV vs 3DIV). However, they didn’t show whether this is regulated transcriptionally or translationally.

– The evidence for a "loop" between Wnt-Shh signalling is thin. The signaling seems to be in one direction: Shh-smo signaling represses Shisa2, and Shisa2 prevents Frizzled3 cell surface expression.

We really appreciate this. It was a misuse of the word. We meant to say “link”. We will remove the “loop” words from the text.

– If Wnts are concluded to be the major the A-P guidance cue, complete loss of anterior turning should be expected but this is not the case, and in fact axons would be predicted to stall at the midline. Additional information, independent of Wnt, might be involved and should be acknowledged, especially with respect to the Wntless knockdown.

We would like to clarify that if Wnts are the main A-P guidance cues, the phenotype should be random turn along the anterior-posterior axis, half turning anterior and the other half turning posterior. This would be the true loss of guidance. Because there are still repulsive cues from the floor plate and the ventral spinal cord, which prevent straight elongation after exiting floor plate or recrossing the midline (Zou et al., 2000), axons turn both anteriorly and posteriorly along the floor plate. If there are other cues for A-P guidance, then axons should still turn anteriorly as the other cues should still work without Wnts. If all axons turn posteriorly, then it means that there are other cues that turn axons posteriorly and Wnts normally overcome that. The fact that axons turn randomly along A-P axis supports the proposal that Wnts are the main (or only) A-P guidance cues. This is what we observed. Our finding that Shh regulate PCP signaling suggests that Shh’s role in A-P guidance is indirect. We would like to mention that after we discovered that Wnts guide axons along the A-P axis in the spinal cord, a number of studies in *C. elegans* showed that Wnts are A-P guidance cues and no other cues were implicated in those cases.

Reviewer # 2:1) The main question is how Shisa2 is regulated by Shh/Smo signaling. It was suggested that this is a gene-expression-based mechanism. However, without direct evidence, it is not appropriate to say it is gene-expression based. As indicated in the discussion, Shisa2 mRNA levels could be regulated by many different mechanisms, and it may not be a Gli target.

Due to the technical limitation (no good commercial antibody to do mouse Gli ChIP experiments), we could not conclude that our findings are dependent or independent of Gli. We will not say it is gene-expression based. Instead, we will say it is Shisa2 mRNA level.

2) The effect of Shisa2 and Shh on Fzd3 glycosylation is interesting. However, the western blot comparing endogenous Fzd3 levels used dorsal spinal cord (Figure 4). Although it reflects the levels of Fzd3 in the cell body of commissural neurons, it may not reflect the levels of Fzd3 in the growth cone. Is it possible to measure the Fzd3 levels in the axon/growth cone of commissural neurons?

Because not only commissural neurons, but also many other cell types are expressed Fzd3, so it is difficult to visualize endogenous Fzd3 in the growth cone in vivo. This is why we could only show the glycosylation level of Fzd3 using dorsal spinal cord extract to show the overall level of Frizzled3 glycosylation.

3) The format of a reference in subsection “Knocking down Shisa2 causes precocious anterior turning of precrossing axons” is wrong.

We appreciate this. We corrected this.

Reviewer # 3Figure 2: It is unclear what the significance is of including the Neuropilin data.

Neuropilin2 was an internal control to show that our RNAseq results matches with in situ results. It is not essential and we removed them.

Figure 3: In the co-immunoprecipitation experiments, better negative controls would increase confidence in the interaction. The only controls included are to show that the antibodies to the tags don't non-specifically pull down the target proteins, Ideally the authors could show that Shisa does not bind to some other membrane protein. (3E) There are insets included in the top row of images, but it's unclear why there are insets, and they're not significantly larger than the boxed area they are enlarging.

We appreciate these suggestions. We tried IRβ, which is not sensitive to Shisa2 expression according to our result, and found that it was not co-immunoprecipitated with Shisa2 (Figure 3). To visualize Fzd3 and Shisa2 localization (in Figure 3), we prepared larger images to show their localization clearly.

Figure 4: (E) the 'changes' in ratio of glycosylated versus unglycosylated Fzd in the Smo knock-outs is not convincing. The different forms are very poorly resolved. It would be interesting if the authors could speculate on how they think Shisa is affecting Fzd glycosylation.

To better evaluate Fzd3 glycosylation in Smo cKO (in Figure 4), we used “Glycoprotein Isolation Kit, WGA” from ThermoFisher to isolate and compare glycosylated Fzd3. And we found that Fzd3 glycosylation was reduced by 90% in Smo cKO.

Figure 5: Shisa2-myc is overexpressed in commissural neurons. Since Shisa2 is myc-tagged, what does shisa-myc expression look like in these neurons? Do the authors observe any difference in Fzd levels or localization in these conditions?

This is a great suggestion. In order to test whether endogenous Shisa2 regulates Fzd3 trafficking in the growth cones of commissural neurons, we utilized the construct in which Frizzled3 was tagged with tdTomato to the carboxyl domain and the FLAG epitope tag was engineered to the extracellular N-terminus (Onishi et al., 2013), and labeled cell surface Fzd3 using anti-Flag antibodies, while total Fzd3 can be detected by tdTomato signal. We electroporated this construct together with Shisa2 shRNA plasmid, and found that more Fzd3 is present on the cell surface when Shisa2 is knocked down. This further supports that Shisa2 indeed suppresses Fzd3 surface presentation in commissural neurons.

Figure 6: Shisa2 is knocked down with shRNA. The authors need to show that this phenotype can be rescued by over-expression of an RNAi-resistant Shisa2 construct.

Two different shRNAs showed very similar phenotype, suggesting that this phenotype is not because of off-target effect. To further eliminate the possibility of off-target effect, we constructed Shisa2 rescue construct, which is insensitive to both shRNA#1 and #2. We then electroporated the shRNA-expressing vector and the rescue vector into the dorsal spinal cord to see how commissural axons turn. We found that Shisa2 rescue construct restored the precocious turning phenotype of Shisa2 knock down, suggesting that Shisa2 knock down phenotype is due to the reduction of Shisa2 expression level.

Figure 5 and Figure 6: Can Frizzled3 protein localization be visualized by immunofluorescence? If yes, what does Frizzled3 localization look like in commissural neurons when Shisa2 is over-expressed, or knocked down?

Because not only commissural neurons, but also many other cell types are expressed Fzd3, so it is difficult to visualize endogenous Fzd3 in the growth cone in vivo. In addition, the endogenous Frizzled3 level is very low. Therefore, we have tried to test exogenous Fzd3 in the commissural neuronal growth cones. However, overexpression of Fzd3 strongly suppressed axon elongation in ex vivo open book explants. All axons could not reach to the floor plate. Therefore, we tested the Frizzled3 localization in Shisa2 overepressing commissural neurons in culture (Figure 6—figure supplement 2). We found that Shisa2 does reduce the cell surface presentation of Frizzled3.

Figure 5—figure supplement 1: a positive control for FGF inhibition is needed.

We appreciate this. Both inhibitors inhibit kinase activity, so we tested whether these can inhibit autophosphorylation of FGFR in our context, and found that only LY2874455 can blocks autophosphorylation (p-Y653/Y654) significantly, so we removed the data about CH-5183284. Please see new Figure 5—figure supplement 1.

[Editors' note: further revisions were requested prior to acceptance, as described below.]

[…] Reviewer 1, on details of experiments in text and figures:1) Figure 2 shows that in Smo mutants there is an increase of Shisa mRNA in a population of Lhx2-low neurons positioned dorsally to the dl1c population. Do these neurons project ipsilaterally? Why do they increase Shisa expression as well?

This is a very good question. These may be later-born Lhx2^high^ dI1 neurons, whose expression of Lhx2 is still not fully activated. In this case, they should have increased Shisa2 level. However, our current goal is to test whether Shisa2 is indeed up regulated in Smo cKO and the Lhx2^high^ dl1 neurons do show higher Shisa2 level. We have now included this discussion in the text.

2) Figure 5 is not very convincing as it is difficult to determine from where the segments of axons indicated by arrows come from. Can the authors replace this image for a more illustrative one?

We thank Reviewer 1 for this suggestion. We have now replaced this image for more illustrative ones that show the crossing and post-crossing segments. These new images have fewer axons and thus are clearer. We should clarify that due to the thickness of the spinal cord open-book explant, we are not able to show the entire trajectory to include the pre-crossing segment, which are buried in the thick gray matter of the ventral spinal cord. But we have confirmed that all these are commissural axons originated from the dorsal spinal cord, an experimental system that my lab established and used in a number of publications.

3) In Figure 6, there are many green/flag puncta outside of the growth cones with the exception of the control one. Can the authors provide better images? Such strong background staining casts doubts on the real increase of surface Fzd3.

We again appreciate this suggestion and provide new images with less background.

4) In Figure 7, there are several axons that seem to project abnormally at the pre-crossing side of the floor plate. This should be explained

This is due to the intrinsic feature of DiI labeling technique as DiI sometimes diffuses to label other classes of neurons, usually more ventral to the dl1 neurons, that turn anterior-posterior before crossing. They tend to have a clear growth pattern and look different from misguided axons, which tend to show inconsistent wondering patterns and turn anterior-posterior randomly. We now explain this in the text.

Reviewer 1 and 3, and the Reviewing Editor, have multiple concerns about the text:1) The manuscript is still extremely poorly written. It suffers from errors of syntax, usage and grammar throughout, and needs to be completely rewritten.

Thank you for the alert. We have now rewritten the text throughout, hoping that it is much improved.

2) There is a pervasive lack scholarly acknowledgement of the literature. Reviewer 3 pointed out in his initial review that the paper is poorly referenced and discussed beyond the authors' own data and their interpretation, yet this has not been fully addressed in the revision:a) The authors argue in point 5 of the response to Reviewer 3's initial comments that the work of Charron, Castellani, and Bashaw did not constitute a genuine gene-expression-based switch mechanism by a morphogen in axon guidance at an intermediate target. We agree with Reviewer 3 and would like to see a more cogent discussion and comparison of the work from these three labs in reference to the present data.

We have now expanded our Discussion to include more completely the work done about the midline guidance and have written a comprehensive review about all the switch mechanisms reported. In paragraph # 2 in the Discussion section, we added 16 new references. This helps to put our work in the context of others.

b) New, very relevant papers on netrin were not cited: two studies have shown that Netrin is actually required in the ventricular zone, not the floor plate and these two studies should be discussed with reference to the authors' findings. Dominici et al., 2017 and Varadarajan et al., 2017. How might these findings interface with the authors' data?

These two new papers about Netrin are not directly related to our paper as our paper does not deal with switch of responsiveness to Netrin. These two Netrin papers highlight the short-range nature of Netrin function, which is very important. The Wnt gradient along the anterior-posterior axis is created by graded levels of gene expression, not by diffusion. Wnts are not very diffusible and commissural axons turn at the border of the floor plate where Wnts are produced. Therefore, the newly found mechanism of Netrin action actually fits very well with how Wnts function (also short-range). We now cite the two new papers as they provide a more precise view about axon guidance mechanisms.

c) While it is true that the manuscript provides novel evidence for a link between Shh signaling and non-canonical Wnt signaling, as Reviewer 1 has pointed out, there are many examples for a similar relationship with canonical Wnt signaling and these should be referenced.

This is a very good point. Urged by the reviewer, we also found a recent paper (2016) that reports a link from Shh to Tbx2/Tbx3 to Shisa3 to canonical Wnt signaling. Therefore, we now point out that our work extends the connection from Shh to PCP (non-canonical signaling), which leads to cellular and tissue morphogenesis, instead of gene expression. This suggests that Shh may use this link to exert precise temporal and spatial control of cell polarity signaling in the cytoplasm that leads to cellular and tissue morphology. We have now revised this part of discussion.

3) The conclusions have been largely tempered in the revised manuscript, yet there are remaining places where the claims have not been toned down, with a number of unclear and unsubstantiated claims given special emphasis. Reviewer 3 finds this problem persistent in the revised manuscript, citing an excerpt from the discussion: "In this study, we also noticed that Shh also up regulates the mRNA level of a Semaphorin receptor Neuropilin2 (Figure 1—figure supplement 2), which may further sensitize commissural axon growth cones to Semaphorin. Therefore, Shh is a switch of growth cone responsiveness to both Wnts and Semaphorins." No experimental evidence is reported in support of this other than the change in Npn expression.

We tried again to tone down conclusion everywhere in the paper. “Shh is a switch of responsiveness to both Wnts and Semaphorins” refers to a previous paper from our lab (Parra and Zou, 2009). In this study, we found that Neuroplilin2 was down regulated in commissural neurons in Smo cKO and we were using Neuropilin2 as our internal control for our RNAseq and in situ hybridization. The activation of the expression regulation of Neuropilin2 is potentially relevant to the role of Shh in activating semophorin and would be consistent with our 2009 paper. However, we semaphoring switch is not the focus of this paper and we do not provide more direct test of the role of Neuropilin2. Therefore, including Neuropilin2 as a control and discussing Neuropilin2 causes confusion. Therefore, we have removed all data about Neuropilin2 in the paper.

References

Sloan, T.F.W., Qasaimeh, M.A., Juncker, D., Yam, P.T., and Charron, F. (2015) Integration of Shallow Gradients of Shh and Netrin-1 Guides Commissural Axons. PLOS Biology *DOI:10.1371/journal.pbio.1002119*

Zhang, X.M., Ramalho-Santos, M. and McMahon A.P. (2001). *Smoothened* Mutants Reveal Redundant Roles for Shh and Ihh Signaling Inducing Regulation of L/R Asymmetry by the Mouse Node. Cell *105*, 781 - 792